# A peptide-neurotensin conjugate that crosses the blood-brain barrier induces pharmacological hypothermia associated with anticonvulsant, neuroprotective, and anti-inflammatory properties following status epilepticus in mice

Lotfi Ferhat[1]*, Rabia Soussi[1], Maxime Masse[2], Grigorios Kyriatzis[1], Stéphane Girard[1,2], Fanny Gassiot[2], Nicolas Gaudin[2], Mathieu Laurencin[2], Anne Bernard[1], Angélique Bôle[1], Géraldine Ferracci[1], Maria Smirnova[3], François Roman[1], Vincent Dive[4], Salvatore Cisternino[3,5], Jamal Temsamani[2], Marion David[2], Pascaline Lécorché[2], Guillaume Jacquot[2], Michel Khrestchatisky[1]*

[1]Aix-Marseille Univ, CNRS, INP, Inst Neurophysiopathol, Marseille, France; [2]VECT-HORUS SAS, Faculté de Médecine, Marseille, France; [3]Université Paris Cité, INSERM UMRS 1144, Optimisation Thérapeutique en Neuropsychopharmacologie, Paris, France; [4]SIMOPRO CEA Saclay, Saclay, France; [5]Pharmacie, Hôpital Universitaire Necker – Enfants Malades, AP-HP, Paris, France

*For correspondence: lotfi.ferhat@univ-amu.fr (LF); michel.khrestchatisky@univ-amu.fr (MK)

## eLife Assessment

The authors developed a method to allow a hypothermic agent, neurotensin, to cross the blood-brain barrier so it could potentially protect the brain from seizures and the adverse effects of seizures. The work is **important** because it is known that cooling the brain can protect it but developing a therapeutic approach based on that knowledge has not been done. The paper is well presented and the data are **convincing**.

**Abstract** Preclinical and clinical studies show that mild to moderate hypothermia is neuroprotective in sudden cardiac arrest, ischemic stroke, perinatal hypoxia/ischemia, traumatic brain injury, and seizures. Induction of hypothermia largely involves physical cooling therapies, which induce several clinical complications, while some molecules have shown to be efficient in pharmacologically induced hypothermia (PIH). Neurotensin (NT), a 13 amino acid neuropeptide that regulates body temperature, interacts with various receptors to mediate its peripheral and central effects. NT induces PIH when administered intracerebrally. However, these effects are not observed if NT is administered peripherally, due to its rapid degradation and poor passage of the blood-brain barrier (BBB). We conjugated NT to peptides that bind the low-density lipoprotein receptor (LDLR) to generate 'vectorized' forms of NT with enhanced BBB permeability. We evaluated their effects in epileptic conditions following peripheral administration. One of these conjugates, VH-N412, displayed improved stability, binding potential to both the LDLR and NTSR-1, rodent/human cross-reactivity and improved brain distribution. In a mouse model of kainate (KA)-induced status epilepticus (SE), VH-N412 elicited rapid hypothermia associated with anticonvulsant effects, potent neuroprotection, and reduced hippocampal inflammation. VH-N412 also reduced sprouting of the

dentate gyrus mossy fibers and preserved learning and memory skills in the treated mice. In cultured hippocampal neurons, VH-N412 displayed temperature-independent neuroprotective properties. To the best of our knowledge, this is the first report describing the successful treatment of SE with PIH. In all, our results show that vectorized NT may elicit different neuroprotection mechanisms mediated by hypothermia and/or by intrinsic neuroprotective properties.

## Introduction

Preclinical and clinical studies have shown that mild to moderate hypothermia is neuroprotective in situations of exacerbated neuronal death including sudden cardiac arrest with resuscitation, ischemic stroke, perinatal hypoxia/ischemia, and traumatic brain injury (TBI) (*Kida et al., 2013*; *Andresen et al., 2015*). Studies also suggest that hypothermia decreases seizure burden in experimental models (*Sartorius and Berger, 1998*; *Schmitt et al., 2006*; *Niquet et al., 2015a*; *Niquet et al., 2015b*) and in humans (*Karkar et al., 2002*; *Kim et al., 2017*). Selective brain cooling has also broad-ranging anti-inflammatory effects and prevents the development of spontaneously occurring seizures in a rat model of post-traumatic epilepsy (*D'Ambrosio et al., 2013*). Results in animal studies are supported by clinical data showing a positive relationship between therapeutic hypothermia (TH) and seizures in neonates with hypoxic-ischemic encephalopathy (*Orbach et al., 2014*). Pediatric case series report treatment of refractory status epilepticus (RSE) with mild hypothermia, which decreases seizure burden during and after pediatric RSE and may prevent RSE relapse. Hypothermia is also used in several centers around the world as second-line therapy for patients with RSE, despite a small evidence base (*Guilliams et al., 2013*; reviewed in *Ferlisi and Shorvon, 2012*; *Bennett et al., 2014*). While the level of hypothermia is uncertain, it has been suggested that mild hypothermia is most effective (*Rossetti and Lowenstein, 2011*). Focal brain cooling also reduces epileptic discharges (EDs) and concentrations of glutamate and glycerol in patients with intractable epilepsy, suggesting neuroprotective effects (*Nomura et al., 2014*). Current methods for the induction of hypothermia largely involve physical cooling therapies, which induce several clinical complications, including electrolyte disturbances, coagulation dysfunction, infections, and cardiac arrhythmia (*Carraway and Leeman, 1975*; *Carraway and Leeman, 1973*). In particular, forced hypothermia lowers core temperature by overwhelming the body's capacity to thermoregulate, but does not change the temperature set point, thus generating counter-regulation mechanisms such as shivering and tremor (*Feketa et al., 2013*; *Suchomelova et al., 2015*), which warrant sedation and curarization in intensive care units (*Andresen et al., 2015*; *Hammer et al., 2009*). Pharmacologically induced hypothermia (PIH) was obtained in animal models using different molecules. They promote a controlled decrease in core temperature by lowering the brain's temperature set point and maintaining thermoregulation at lower set points (*Liska et al., 2018*). Among those, neurotensin (NT) is a 13 amino acid neuropeptide that modulates body temperature (*Coquerel et al., 1988*; *Coquerel et al., 1986*; *Fanelli et al., 2015*). NT interacts with three receptor subtypes, including NTSR1, NTSR2, and gp95/Sort-1 or NTSR3, to mediate its peripheral and central effects. The G protein-coupled receptors NTSR1 and NTSR2 have seven transmembrane domains (*Vincent, 1995*) while Sort1/NTSR3 only has one single transmembrane domain and is not coupled to a G protein (*Mazella, 2001*). The role of NT in neuroprotection and neuroinflammation, and the receptors involved, remain largely unknown. NTSR1 and NTSR2 differ in their affinity for NT, with NTSR1 and NTSR2 showing high and lower affinity, respectively (*Tanaka et al., 1990*; *Chalon et al., 1996*). NTSR1 is expressed prenatally, preferentially in neurons in different brain structures (*Palacios et al., 1988*), while NTSR2 is expressed postnatally, essentially in glial and endothelial cells and increases during brain development (*Sarret et al., 1998*; *Lépée-Lorgeoux et al., 1999*; *Yamauchi et al., 2007*; *Woodworth et al., 2018*; *Kyriatzis et al., 2021*).

NT has been shown to induce PIH when administered intracerebrally (*Coquerel et al., 1986*; *Coquerel et al., 1988*; *Popp et al., 2007*; *Fanelli et al., 2015*), by inducing a downward shift of the physiological temperature set point (*Gordon et al., 2003*). However, these effects are not observed if NT is administered peripherally due to its rapid processing by peptidases and poor passage of the blood-brain barrier (BBB). A number of NT analogues have been generated that are more stable than NT and that cross the BBB to induce hypothermia (reviewed in *McMahon et al., 2002*; *Gordon et al., 2003*; *Orwig et al., 2009*; *Boules et al., 2013*). These analogues have shown significant neuroprotection in several models of acute brain damage such as hypoxic ischemia, stroke, and TBI (*Choi et al.,*

*2012*; *Gu et al., 2015*; *Zhong et al., 2020*). However, to our knowledge, there are no reports on the effects of PIH in EDs. One of our main objectives was to assess such effects in experimental epileptic conditions. For this purpose, we generated 'vectorized' forms of NT that cross the BBB and that display potent hypothermic properties. Indeed, transport of active principles across the BBB can be enhanced by conjugation to ligand or 'vector' molecules designed to bind specific receptors involved in receptor-mediated transcytosis (RMT) (*Pardridge, 2001*; *Pardridge, 2003*; *de Boer and Gaillard, 2007*; *Jones and Shusta, 2007*; *Pardridge, 2007*; reviewed in *Vlieghe and Khrestchatisky, 2013*). Several BBB receptors have been described that undergo RMT, including the transferrin receptor, the insulin receptor, the insulin-like growth factor receptor, and receptors of the low-density lipoprotein receptor (LDLR) family. A number of antibodies, protein ligands, or peptides that bind some of these receptors have been developed as vectors to carry pharmacological payloads across the BBB (*Friden et al., 1991*; *Wu et al., 1997*; *Wu and Pardridge, 1998*; *Boado et al., 2007*; *Pan et al., 2004*; *Spencer and Verma, 2007*).

LDLR is part of a group of single transmembrane glycoproteins, referred to as cell surface endocytic receptors. They bind apolipoprotein complexes and are expressed with some degree of tissue specificity (*Brown and Goldstein, 1979*; *Herz and Bock, 2002*). We previously described the rational characterization and optimization of a family of cyclic peptides that bind the LDLR in vitro and in vivo. These peptides bind the EGF-precursor homology domain of the LDLR and thus do not compete with LDL binding on the ligand-binding domain. To our knowledge, they have no beneficial or untoward effects on LDL binding and LDLR activity (*Malcor et al., 2012*; *Jacquot et al., 2016*; *David et al., 2018*; *Varini et al., 2019*; *Acier et al., 2021*; *Yang et al., 2023*; *Broc et al., 2024*). These peptides can transport across the BBB and into the CNS and specific organs, different payloads in an LDLR-dependent manner. We and others have shown that such payloads include fluorophores, proteins, nanoparticles, and liposome-based cargos (*Malcor et al., 2012*; *Zhang et al., 2013*; *Chen et al., 2017*; *Molino et al., 2017*; *Cui et al., 2018*; *David et al., 2018*; *Shen et al., 2018*). In the present work, we conjugated several of our LDLR-targeting peptides to NT and to shorter active variants of NT (residues 6–13 and 8–13) with different linkers. These conjugates displayed binding potential to both the LDLR and NTSR-1 receptors, with rodent/human cross-reactivity, enhanced metabolic stability in plasma compared to the native NT, and improved brain penetration potential. We selected the VH-N412 conjugate for further studies in mouse, owing to its potent hypothermia following intravenous (i.v.) administration at low dose together with optimal chemistry and conjugation features. The properties of VH-N412 were evaluated in a mouse model of kainate (KA)-induced SE, a model of seizures associated with neurodegeneration, neuroinflammation, and network reorganization. Following induction of SE, we show that the VH-N412 compound elicited rapid hypothermia that was associated with anticonvulsant effects. Seven days following SE, we observed potent neuroprotection and reduced inflammation in the hippocampus, a structure highly vulnerable to damage at early stages of epilepsy. Neuroprotection elicited by VH-N412 also reduced significantly aberrant sprouting of the dentate gyrus (DG) mossy fibers assessed 2 months after SE and preserved learning and memory skills in the treated mice. We showed that NTSR1, one of the NT receptors, was expressed in hippocampal pyramidal neurons in vitro and in vivo, in cell bodies, dendrites, and spines, the postsynaptic compartment of glutamatergic synapses. Besides the neuroprotective hypothermia effects observed in vivo with VH-N412, we show in cultured hippocampal neurons challenged with neurotoxic NMDA or KA glutamate agonists that VH-N412 displayed temperature-independent neuroprotective properties that are as potent as oestradiol or BDNF.

## Results

### Synthesis and purification of peptide-NT conjugates and their hypothermic potential in mice

We conjugated the 8-mer VH445 cyclic peptide vector that binds the LDLR (peptide 22: [cMPRL-RGC]$_c$) (*Malcor et al., 2012*) to the NT tridecapeptide through its lysine in position 6 using a three-step reaction sequence. Both peptides were prepared by solid-phase peptide synthesis on a CEM Liberty microwave peptide synthesizer using standard Fmoc/tert-butyl chemistry. Cyclization of the VH445 peptide was performed on crude peptides (*Malcor et al., 2012*) by formation of a disulfide bridge between the two VH445 cysteine residues. $K_3[Fe(CN)_6]$ was used as an oxidating reagent. A

sulfo-*N*-[ε-maleimidocaproyloxy]succinimide ester (sulfo-EMCS) was used to incorporate a maleimido hexanoic acid linker (MHA) at lysine 6 of NT, resulting in [Lys(MHA)6]NT. In parallel, a reactive thiol moiety was incorporated on the VH445-G modified peptide by derivatization of the acid C-terminal (C-ter) with cysteamine. Conjugation was performed between both functionalized intermediates [Lys(MHA)6]NT and VH445-G-$(CH_2)_2$-SH leading to conjugate VH-N21 (*Figure 1A*). VH-N21 and NT were administered i.v. (bolus) in the tail vein of Swiss CD-1 mice and body temperature was monitored using digital thermometer rectal probes at different time points following conjugate administration. VH-N21 induced a transient and mild hypothermia, that was dose-dependent and that reached a maximum of –3.7°C 1 hr after injection, with a dose of 8 mg/kg molar equivalent NT (eq. NT). In contrast, no significant hypothermia was observed with native NT at the same dose (*Figure 1B*). We next generated other peptide-NT conjugates based on peptides with improved properties in terms of stability and binding to the LDLR (*Jacquot et al., 2016*; *David et al., 2018*). In particular, the VH4129 peptide ([cM"Pip"RLR"Sar"C]$_c$) was chosen for its optimal stability/binding properties (*Jacquot et al., 2016*). The plasma metabolic stability of VH4129 was shown to be higher than that of VH445 ($t_{1/2}$ 7 hr vs 3 hr, respectively), owing to a rationally optimized insertion of non-natural amino acids. When compared with VH445, the LDLR binding of VH4129 was overall similar ($K_D$ 60–70 nM). However, compared with the VH445 peptide, VH4129 presented the highest association rate (19.2 vs 7.6×10⁵ s⁻¹·M⁻¹ for VH4129 and VH445, respectively) and the highest dissociation rate (12.2 vs 5.9×10⁻² s⁻¹ for VH4129 and VH445, respectively). These properties provided excellent potential for peptide binding to BBB-exposed LDLR while allowing efficient release in the parenchymal compartment. Consistently, using the same initial thiol-maleimide coupling strategy as for the VH-N21 conjugate, the resulting VH-N41 conjugate (*Figure 1C*) induced a stronger and more sustained hypothermia after i.v. injection in mice, with a maximal body temperature decrease of –6.8°C (*Figure 1D*). We next evaluated a series of conjugates allowing one-pot linear synthesis with different versions of NT (residues 2–13, 6–13, and 8–13), while spacing the LDLR-targeting peptide from the NT peptide using linkers such as the glycine tripeptide (GGG), aminohexanoic acid (Ahx), or polyethylene glycol (PEG6). In this strategy, full size conjugates were synthesized in a one-step procedure on a CEM Liberty microwave peptide synthesizer using standard Fmoc/tert-butyl chemistry except for conjugates containing a PEG6 linker that was introduced manually. Cyclization was performed on crude peptides with the same procedure as described for peptide vectors. With this new strategy synthesis yields were significantly increased compared to our initial thiol-maleimide conjugation strategy (*Supplementary file 1*). The conjugates were all evaluated for their potential to induce hypothermia in mice (*Supplementary file 2*) and allowed the selection for further studies of the VH-N412 conjugate that encompasses a PEG6 linker between the VH4129 peptide and NT(8–13) (*Figure 1E*). Importantly, this VH-N412 conjugate elicited a hypothermic response similar to that of VH-N41, with a maximal body temperature decrease of –6.4°C, but with an even more sustained profile (*Figure 1F*). No effect was observed with the control PEG6-NT(8–13) compound, confirming the involvement of the VH4129 peptide in the hypothermic effect of VH-N412. Dose-response curves confirmed that VH-N412 displayed an ED50 similar to that of VH-N41, estimated at 0.69 and 0.93 eq. NT, respectively (corresponding to 0.80 and 1.67 mg/kg, respectively) (*Figure 1G*). With its smaller size and easier production using one-pot linear synthesis, thereby leading to higher synthesis yields, VH-N412 was selected for further investigation in the mouse model of KA-induced SE.

## In vitro and in vivo biological properties of VH-N412: LDLR and NTSR1 binding, plasma stability, and BBB transport

In parallel with the in vivo selection of VH-N412, we verified in vitro that conjugation of its VH4129 peptide and NT(8–13) moieties did not interfere with its potential to bind the LDLR. This was confirmed using surface plasmon resonance (SPR) on immobilized human LDLR, with free VH4129 and the VH-N412 conjugate displaying very similar binding affinity and profiles, namely 72.6 and 63.8 nM, respectively (*Figure 2A*). We also studied the binding properties of the VH-N412 conjugate to NTSR-1. Both native NT(1–13) and VH-N412 were assessed for binding competition with a reference radiolabeled NT on cell membrane extracts expressing either the rat or the human form of NTSR-1. Both NT(1–13) and VH-N412 showed similar Ki values for rNTSR-1 and hNTSR-1, in the low nanomolar range (*Figure 2B*, right and left graphs). Next, the proteolytic resistance of VH-N412 was evaluated and compared to the native NT as well as the initial VH-N21 conjugate by incubation

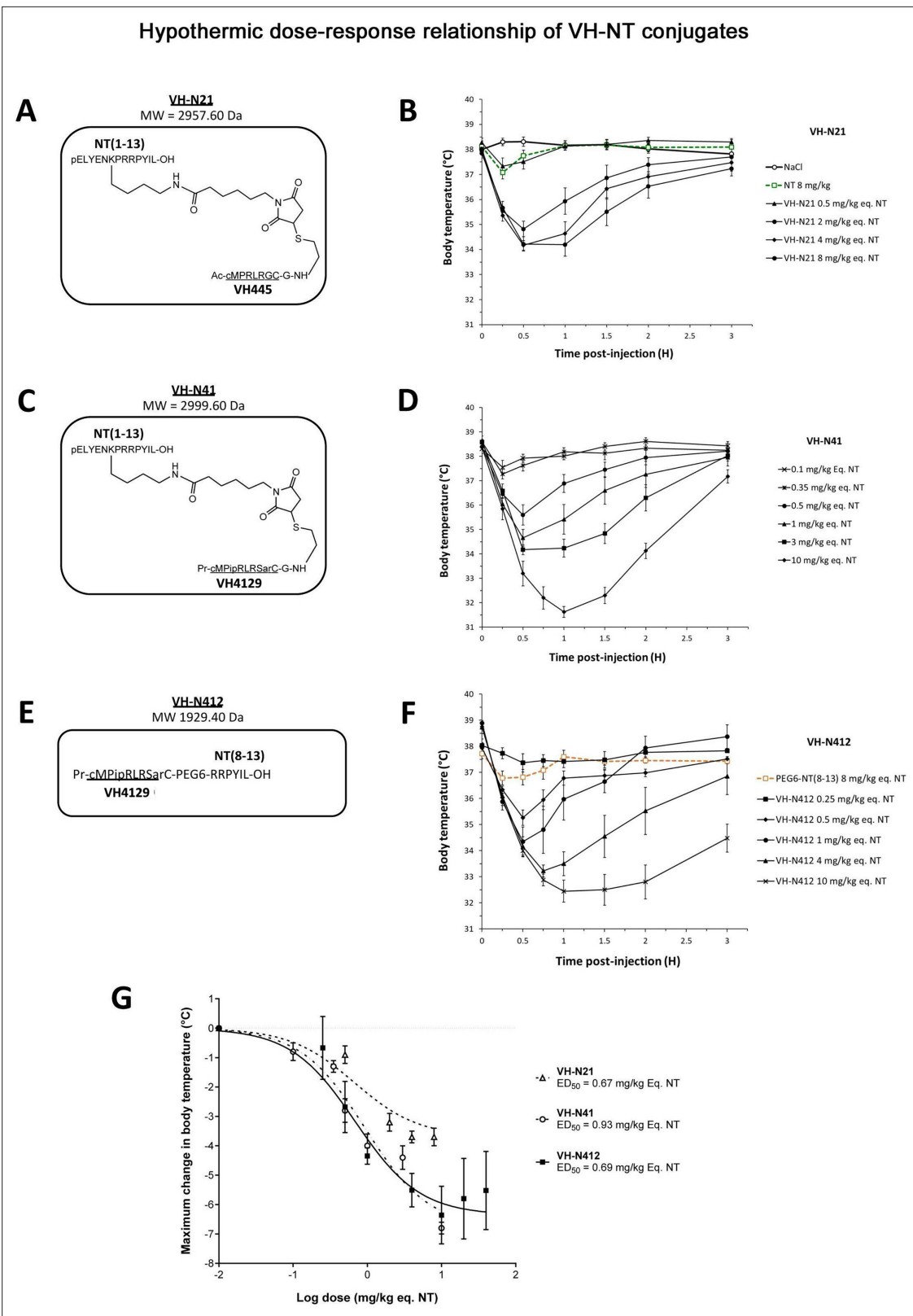

**Figure 1.** Hypothermic dose-response relationship of different VH-NT conjugates following single intravenous (i.v.) (bolus) injection in naïve Swiss (CD-1) mice. (**A, C, and E**) Chemical structure and molecular weight of the VH-N21, VH-N41, and VH-N412 conjugates, containing the eight amino acid cyclic brain penetrating peptide that recognizes the LDLR (VH445 for VH-N21 and VH4129 for VH-N41 and VH-N412), and either the neurotensin (NT) tridecapeptide (VH-N21 and VH-N41) or its C-terminal NT(8–13) fragment (VH-N412). (**B, D, and F**) Hypothermic response to VH-N21, VH-N41, and

*Figure 1 continued on next page*

Figure 1 continued

VH-N412 conjugates in mice after single i.v. (bolus) injection at increasing dose levels. Core body (rectal) temperature was measured before (baseline) and at indicated times after injection. Data are presented as means ± SEM, n=4–8 per group. (**G**) Dose-response curves of VH-N21, VH-N41, and VH-N412 hypothermic response. $ED_{50}$ values for each conjugate were estimated by plotting the response vs log[dose(mg/kg eq. NT)] followed by nonlinear regression (three parameters) using GraphPad Prism software.

in freshly collected mouse blood at 37°C followed by quantification of the parent compound in the plasma fraction using LC/MS-MS. As opposed to the very low resistance of the native NT ($t_{1/2}$ of 9 min), both VH-NT conjugates showed greatly enhanced stability, with VH-N412 showing the highest in vitro half-life estimated at 83 min, compared to 44 min with VH-N21 (*Figure 2C*). The in vitro half-life of VH-N412 was estimated at 74 min in human blood, demonstrating similar metabolic resistance across species (*Figure 2D*).

The mouse in situ brain perfusion method described by *Dagenais et al., 2000* was used to measure BBB transport rate clearance ($K_{in}$) of tritium-labeled NT, VH-N21, and VH-N412 conjugates. Consistent with the previously reported NT $K_{in}$ of 0.013 μL/s/g measured in mice (*Gevaert et al., 2016*), NT demonstrated a very low BBB transport, with a $K_{in}$ of ~0.04 μL/s/g of brain tissue (*Figure 2E*). In contrast, VH-N21 and VH-N412 showed $K_{in}$ values of 0.13 and 0.37 μL/s/g respectively (*Figure 2E*), demonstrating that conjugating NT with these peptide vectors enhanced its BBB transport. Furthermore, VH-N412 did not alter the integrity of the BBB. Indeed, the brain distribution volume of $^{14}C$-sucrose as a marker of brain vascular volume in VH-N412 mice (19.00±1.00 μL/g) was in the normal range (i.e. Vvasc <20.00 μL/g) (*Cattelotte et al., 2008*) and similar to that of NT (18.00±2.00 μL/g). Although we confirmed that the LDLR peptides we developed do not bind to LRP-1 or LRP-8 receptors (not shown), we cannot exclude that they could nevertheless bind to some extent to the EGF-precursor homology domains of other receptors of the LDLR family or to other proteins encompassing this domain and expressed at the BBB.

Taken together, our results demonstrate that the VH-N412 conjugate retains its binding potential to both the LDLR and NTSR-1 receptors, with rodent/human cross-reactivity. VH-N412 encompasses the VH4129 peptide vector with higher association and dissociation rates to LDLR compared to VH445. VH-N412 displayed greatly enhanced metabolic stability in plasma compared to the native NT, but also to the initial conjugate VH-N21, and displayed higher $K_{in}$ properties with sharp improvement of brain penetration potential compared to VH-N21. These combined features contribute to the high hypothermic potential of VH-N412, requiring plasma resistance, improved BBB permeability, and potent binding to its pharmacological target, namely brain NTSR1. Thus, VH-N412 appeared as an ideal candidate for further investigation of its central pharmacological potential in pathophysiological situations in vivo. Finally, tolerability studies were performed in naïve mice with the administration of up to 20 and 40 mg/kg eq. NT (i.e. 25.8 and 51.6 mg/kg of VH-N412) with n=3 for these doses. The rectal temperature of the animals did not fall below 32.5 to 33.2°C, similar to the temperature induced with the 4 mg/kg eq. NT dose. We observed no mortality or notable clinical signs other than those associated with the rapid HT effect such as a decrease in locomotor activity. We thus report a very interesting therapeutic index since the maximal tolerated dose was >40 mg/kg eq. NT, while the maximum effect is observed at a 10× lower dose of 4 mg/kg eq. NT and an ED50 established at 0.69 mg/kg as shown in *Figure 1G*. Severe hypothermia could also be induced in rats with different conjugates similar to VH-N412 with the same efficacy and safety (data not shown).

## Effect of VH-N412 in a model of KA-induced seizures

We assessed our VH-N412 conjugate in a model of KA-induced seizures using adult male FVB/N mice. This mouse strain was selected as a reliable and well-described mouse model of epilepsy, where seizures are associated with cell death and neuroinflammation (*Schauwecker, 2003*; *Wu et al., 2021*). KA was administered subcutaneously (s.c.) in FVB/N mice at the dose of 45 mg/kg. The scheme in *Figure 3A* shows the timeline of the experiments we performed, including physiological, histopathological, behavioral, and synaptogenesis assessment. Five groups of mice were generated: SHAM, SE, SE+VH-N412, SE+NT(8–13), SE+diazepam (DZP). Body temperature was monitored using a rectal probe before KA injection and every 30 min during 2.5 hr thereafter. SE occurred around 2 hr after KA injection (KA-2H) and was characterized by stage 5–6 seizures and often associated with some hyperthermia (nonsignificant, *Figure 3B*; *Supplementary file 3*). VH-N412 administered at the dose

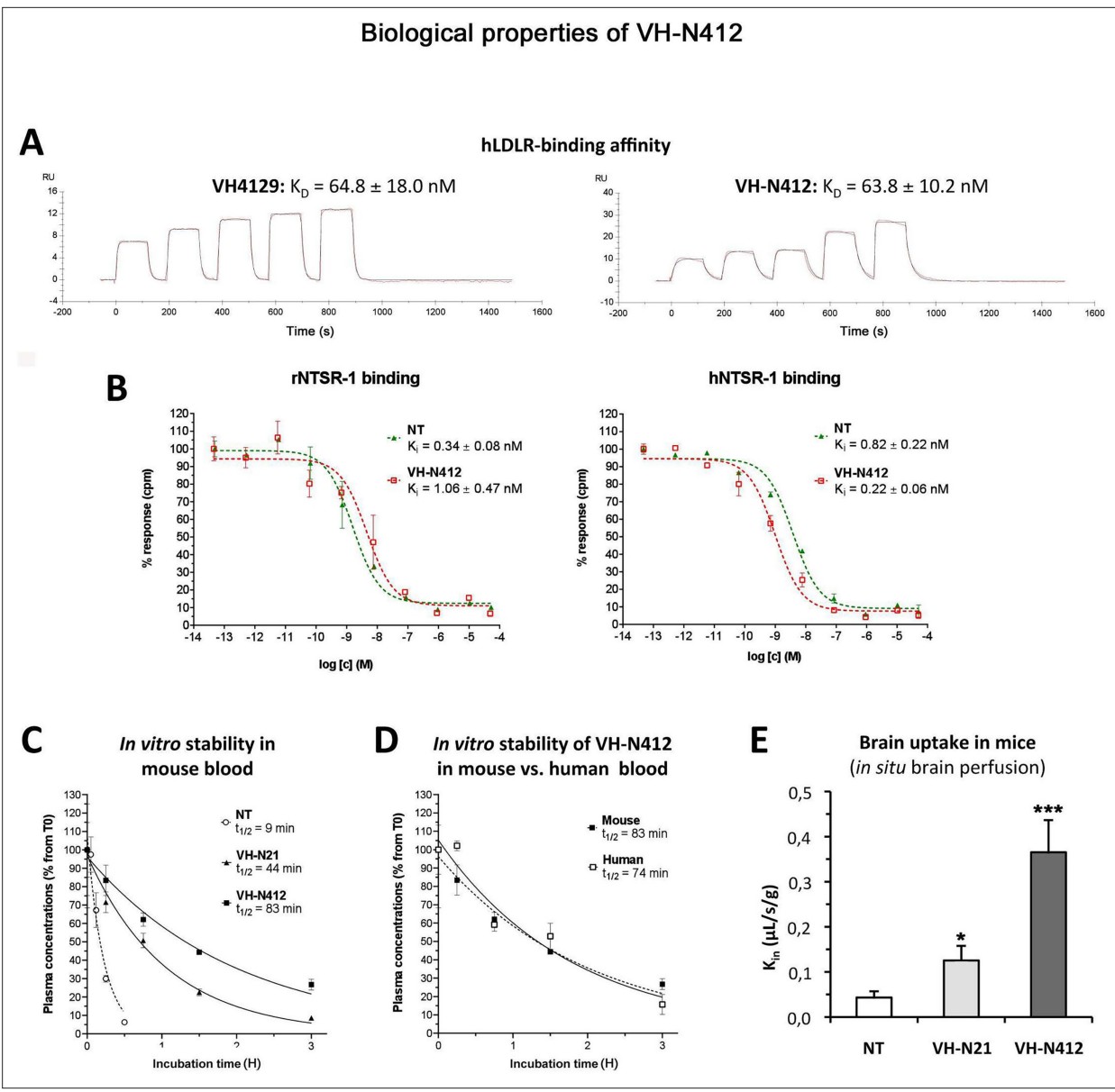

**Figure 2.** In vitro and in vivo biological properties of VH-N412. (**A**) Surface plasmon resonance (SPR) sensorgrams of the free VH4129 and the VH-N412 compound on immobilized human LDLR (hLDLR). Red lines show the specific binding of molecules obtained after double subtraction of the signal measured on the control flow cell (without immobilized LDLR) and a blank run. Black lines show fit curves of the experimental data with a 1:1 binding model. The illustrated data are representative of two to five independent experiments. (**B**) Dose-response inhibition curves of tritiated NT, bound on hNTSR-1 or rNTSR-1 membrane extracts, in the presence of indicated concentrations of NT or VH-N412. Indicated Ki values were estimated from mean IC50 values obtained by logarithmic regression of experimental data. Data were plotted as means ± SD of biological duplicates. (**C and D**) Comparison of degradation rates for NT or peptide-NT conjugates in mouse blood. NT or peptide-NT conjugates were incubated in freshly collected mouse (**C**) or human (**D**) blood and analyzed using liquid chromatography-tandem mass spectrometry (LC-MS/MS) at indicated times in the plasma fraction. Data were plotted as means ± SD of n=3 biological replicates. $T_{1/2}$ values were estimated from nonlinear regression (one-phase decay) of experimental data. (**E**) Blood-brain barrier (BBB) transport of tritium-labeled NT or peptide-NT conjugates using in situ brain perfusion in mice. Data were presented as mean ± SEM for three to six animals. Student's t-test vs NT: *p<0.05, **p<0.01.

of 4 mg/kg eq. NT 30 min after SE onset (SE30), hence 2.5 hr after s.c. administration of KA (KA-2H), invariably led to transient hypothermia (*Figure 3B*; *Supplementary file 3*), which persisted at least 2 hr. Mean decreases in body temperature of –2.12°C were recorded at SE30 for SE+VH-N412 animals (36.50 ± 0.34°C, p<0.01, Tukey's test) as compared to SE animals (38.00 ± 0.37°C) (*Figure 3B*; *Supplementary file 3*). This hypothermia was associated with a significant decrease of seizures in the SE+VH-N412 group (1.97±0.36, p<0.01, Tukey's test) at SE30 as compared with the SE group (5.38±0.15)

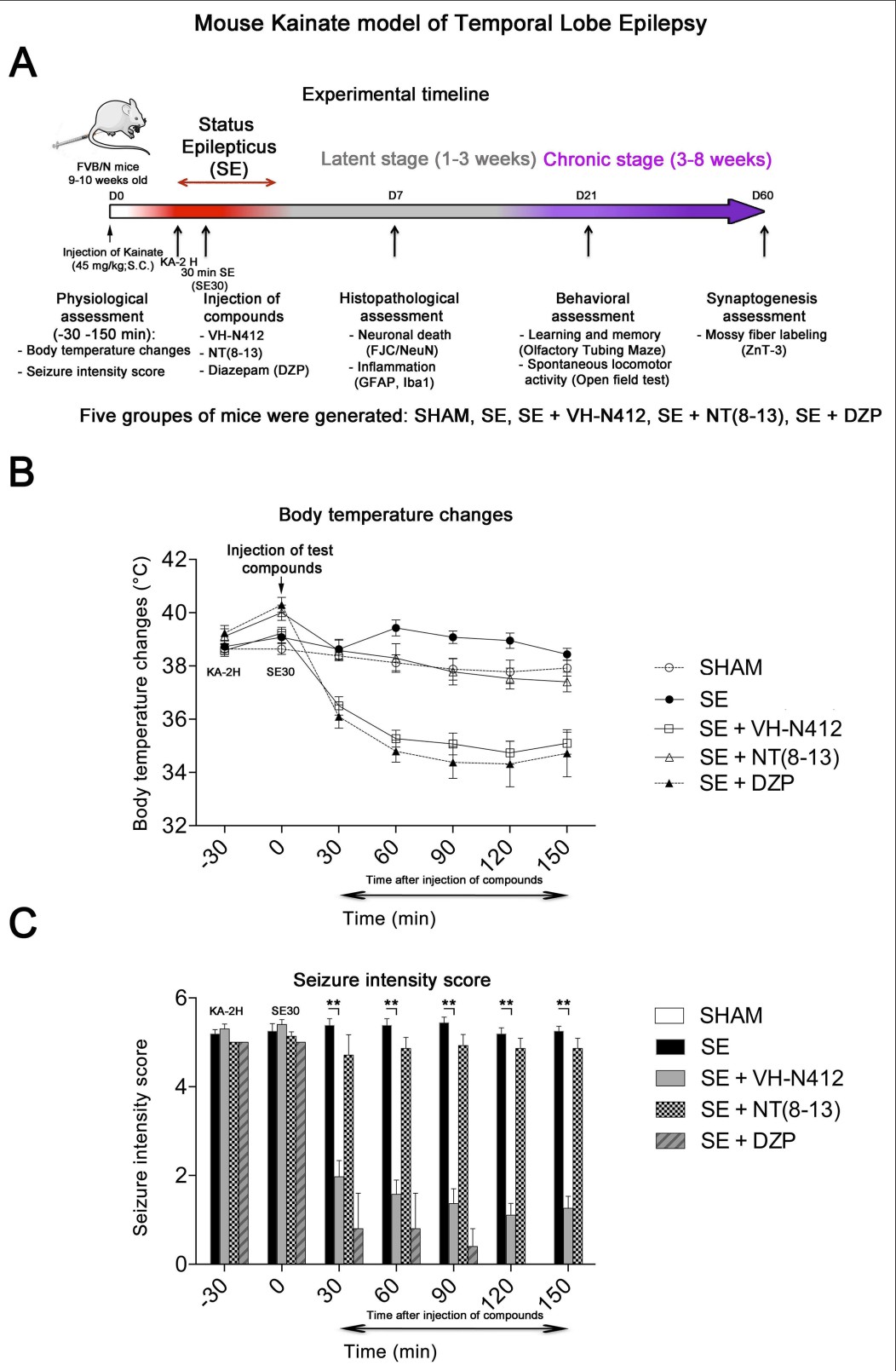

**Figure 3.** Effects of VH-N412 on body temperature and seizure intensity following status epilepticus (SE). (**A**) Experimental timeline with the assessment of physiological, histopathological, behavioral, and synaptogenesis features associated with the mouse KA model of temporal lobe epilepsy. Five groups of mice were generated: SHAM, SE, SE+VH-N412, SE+NT(8–13), SE+DZP. (**B**) Mice were injected with KA, which induced stage 5 or stage

*Figure 3 continued on next page*

*Figure 3 continued*

6 seizures after 2 hr, characteristic of SE, associated with hyperthermia as compared to all animal groups. VH-N412 was administered 30 min after SE onset at the dose of 4 mg/kg eq. NT caused significant hypothermia, which persisted at least 2 hr, similar to the effects of high-dose DZP (45 mg/kg) administered i.p. and used as positive control. SE+NT(8–13) had no effect on body temperature when administered 30 min after SE onset. (**C**) Hypothermia induced by VH-N412 was associated with a significant decrease of seizures in the SE+VH-N412 group, similar to DZP, while SE+NT(8–13) had no effect on seizure intensity.

---

(*Figure 3C*; *Supplementary file 4*). SE+VH-N412 animals presented an average of seizure intensity score of 2 or less during the rest of the experiment (SE30-SE150 min). A subset of animals was administered i.p. a high dose of DZP (15 mg/kg), used as a positive control for its anticonvulsant effects in seizure models (for review, see *Sharma et al., 2018*) and its hypothermic effects (*Vinkers et al., 2009*). As for SE+VH-N412 animals (average seizure intensity score >5 at SE30), SE+DZP animals rapidly showed an average seizure intensity score of 1–2 during the rest of experiments, and significant hypothermia was also observed in these animals at all time points (*Figure 3B and C*; *Supplementary file 3 and 4*; p<0.01, Tukey's test). No significant variations of body temperature or seizure intensity score were observed when NT8–13 was administered at SE30, as compared with SE animals at all time points (*Figure 3B and C*; *Supplementary files 3 and 4*).

## Effects of VH-N412 on neurodegeneration and inflammation in hippocampus

Rodents that experienced SE developed inflammation and lesions in several brain areas. In our study, we focused on the hippocampal formation, where inflammation and cell death occur within the first days following KA-induced SE (*Gröticke et al., 2008*; *Lévesque and Avoli, 2013*; *Li and Liu, 2019*).

### Effects of VH-N412 on hippocampal neurodegeneration

The effects of VH-N412 on hippocampal neural cell degeneration were assessed 7 days after SE (*Figure 4A*) using Fluoro-Jade C (FJC) staining in SHAM, SE, SE+VH-N412, SE+NT(8–13), SE+DZP animals (*Figure 4A*). Representative photomicrographs of hippocampal pyramidal cells from Cornu Ammonis areas 1 (CA1), 3 (CA3), Hilus (H), and granule cell layer (GCL) areas that were quantified are shown in *Figure 4B* while high magnification of these same areas stained with FJC in all groups of animals are shown in *Figure 5*. Semiquantitative analysis revealed that FJC staining is significantly increased in CA1 ($222.73 \pm 11.80\%$, 123%, p<0.01, Tukey's test), CA3 ($197.27 \pm 14.58\%$, 97%, p<0.01, Tukey's test), and to a lesser extent in H ($120 \pm 4.98\%$, 20%, p<0.05, Tukey's test) of SE animals compared to SHAM animals (CA1: $100 \pm 2.42\%$; CA3: $100 \pm 3.18\%$; H: $100 \pm 4.62\%$). No difference in FJC staining was found in the GCL ($100 \pm 5.44\%$; n=3 mice; p>0.05; ANOVA) (*Figure 4C*). These results indicate that there is major neural cell death in all hippocampal layers, including CA1–3 pyramidal cell layers and H of the DG. Neural cell degeneration observed in SE animals was significantly decreased when VH-N412 was administered at SE30 (CA1: $97.97 \pm 2.60\%$; CA3: $109.58 \pm 7.31\%$; H: $89.82 \pm 3.20\%$; p<0.01, Tukey's test) (*Figure 4C*). In contrast, no changes were observed when NT8-13 was administered (CA1: $222.59 \pm 12.02\%$; CA3: $200.08 \pm 11.84\%$; H: $100.98 \pm 5.75\%$; p>0.05; ANOVA) (*Figure 4C*). A subset of animals was administered i.p. with a high dose of DZP (15 mg/kg) used as a positive control for its neuroprotective effects in seizure models. The results obtained for SE+VH-N412 animals were not different from those observed in SE+DZP mice (CA1: $99.21 \pm 2.88\%$; CA3: $106.12 \pm 3.56\%$; H: $102.07 \pm 2.79\%$; p>0.05; ANOVA) (*Figure 4C*). Immunohistochemistry for the neuronal nuclear antigen (NeuN) was also performed to confirm neuronal degeneration, and to evaluate on NeuN and FJC sections the effects of VH-N412 in animals at 7 days post SE. Representative photomicrographs of neurodegeneration in SE animals (compare SHAM vs SE) and neuroprotection mediated by VH-N412 are shown (*Figure 6*, in green, left panels). SE animals displayed a decreased NeuN staining and significantly increased neuronal death score in the hippocampal formation compared to SHAM animals. When VH-N412 or DZP was administered at SE30, the neuronal death score was significantly reduced by 51% and 34% respectively (p<0.01, Tukey's test; *Figure 6B*, left histogram). However, no changes were observed when NT8-13 was administered (p>0.05; ANOVA) (*Figure 6B*, left histogram). Altogether, these results indicated that SE-induced neurodegeneration was partially prevented by VH-N412.

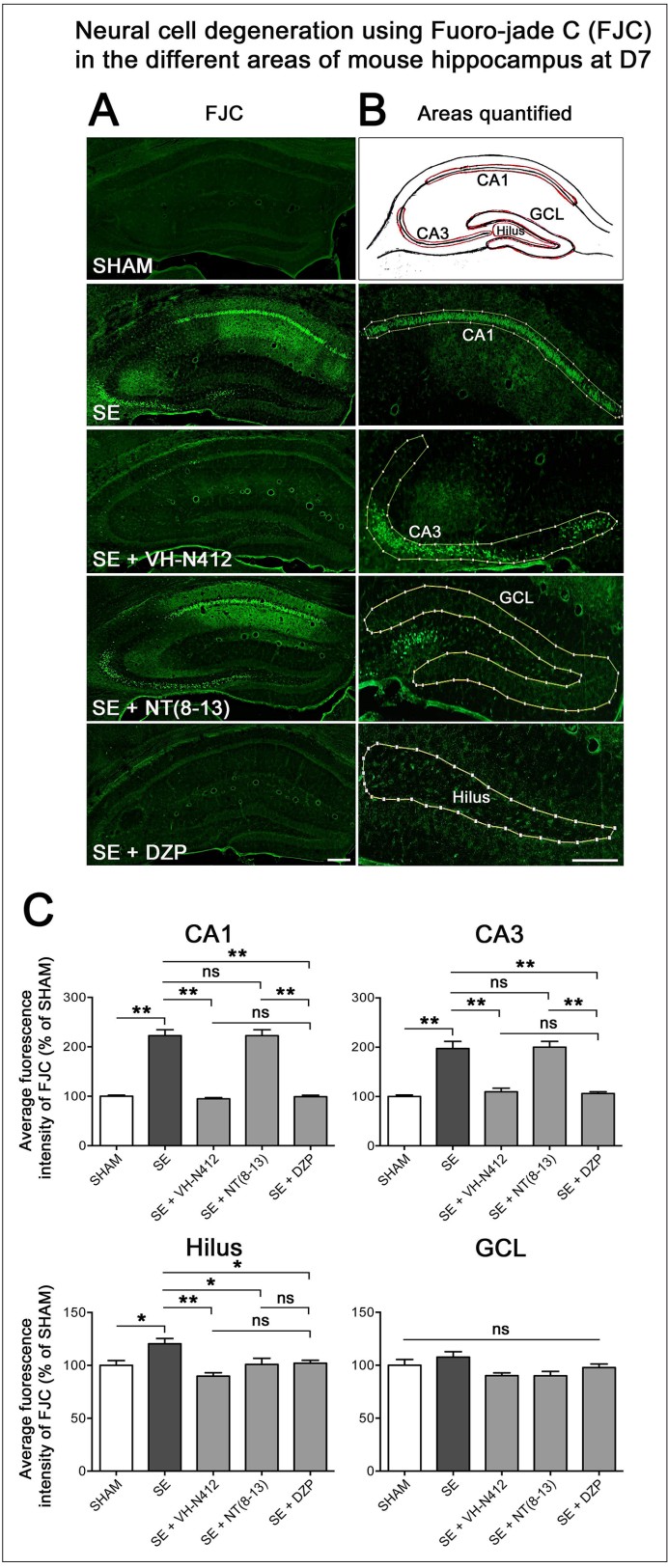

**Figure 4.** Effects of VH-N412 on neural cell degeneration following KA-induced status epilepticus (SE). (**A**) Fluoro-Jade C (FJC) staining was used to assess the extent of neural cell damage in coronal sections of the dorsal hippocampal formation at D7 post-SE from SHAM, SE, SE+VH-N412, SE+ NT(8–13) and SE+ DZP animals. (**B**) The regions of interest were highlighted on the scheme, upper panel, and were traced to quantify FJC in the CA1,

*Figure 4 continued on next page*

*Figure 4 continued*

CA3, GCL, and the H. Scale bars: 200 μm in all panels. (**C**) Histograms comparing the mean intensities of staining for FJC in dorsal CA1, CA3, H, and GCL from SHAM, SE, SE+VH-N412, SE+NT(8–13), and SE+DZP animals. VH-N412 as well as DZP displayed significant protective effect in dorsal CA1, CA3, and H but not in GCL. Data were expressed as the average percentage ± SEM, normalized to the SHAM CTL. Asterisks indicate statistically significant differences: *p<0.05, **p<0.01 (Tukey's test).

## Effects of VH-N412 on glial-mediated inflammatory response

We used glial fibrillary acidic protein (GFAP)- and ionized calcium-binding adaptor molecule 1 (Iba1)-immunolabeling to evaluate the effects of VH-N412 on astroglial (*Figure 6A*, in green, middle panels) and microglial (*Figure 6A*, in red, right panels) reactivity respectively. In SHAM animals, basal labeling for GFAP (100 ± 20.77%) and Iba1 (100 ± 18%) was visible in the hippocampal formation (*Figure 6B*, middle histogram). In SE animals, a significant activation of glial cells occurred in all hippocampal areas (GFAP: 277.74 ± 37.96%, 178%; p<0.01, Tukey's test; *Figure 6B*, middle histogram; Iba1: 440.04 ± 29.86%, 340%; p<0.01, Tukey's test; *Figure 6B*, right histogram). This inflammatory response was significantly decreased when VH-N412 was administered at SE30 (GFAP: 96.44 ± 4.48%; 181.50%; p<0.01, Tukey's test; *Figure 6B*, middle histogram; Iba1: 257.69 ± 16.96%, 182.31%; p<0.01, Tukey's test; *Figure 6B*, right histogram) or DZP (GFAP: 56.32 ± 6.61%; 221.41%; p<0.01, Tukey's test; *Figure 6B*, middle histogram; Iba1: 164.59 ± 4.88%, 275.41%; p<0.01, Tukey's test; *Figure 6B*, right histogram). No significant changes were observed when NT(8–13) was administered at SE30 (GFAP: 316.87 ± 24.36%; p>0.05; ANOVA; *Figure 6B*, middle histogram; Iba1: 383.32 ± 40.40%; p>0.05; ANOVA; *Figure 6B*, right histogram). Altogether, these results indicated that SE-induced neuroinflammation was partially prevented by VH-N412 treatment.

## Effects of VH-N412 on mossy fiber sprouting

Temporal lobe epilepsy is associated with sprouting of the mossy fibers in the inner molecular layer (IML), in response to hilar cell loss (*Jiao and Nadler, 2007*; *Sloviter et al., 2006*). To further investigate the neuroprotective effect of VH-N412, we evaluated the extent of mossy fiber sprouting 8 weeks after SE. Since mossy fiber terminals are highly enriched in zinc ions, we used immunohistochemical labeling for the zinc vesicular transporter 3 (ZnT-3) to detect mossy fiber sprouting as illustrated in *Figure 7*. In all SHAM animals, mossy fiber terminals were present in the hilus region and no terminals were observed in the GCL and IML of the DG (*Figure 7A*). In SE animals, mossy fiber terminals were not only observed in the hilar region as in SHAM mice, but also within the IML and GCL of the DG (*Figure 7A*). In comparison with SE animals (50.38±15.57), the number of terminals innervating the IML was significantly reduced in animals administered with VH-N412 at SE30 (7.79±2.54, 84.53%; p<0.01, Tukey's test, *Figure 7B*) but was not significantly different when NT(8–13) (25.85±6.39; p>0.05; ANOVA) was administered at SE30 (*Figure 7B*). Altogether, these results indicated that SE-induced mossy fiber sprouting was partially prevented by VH-N412 treatment.

## **Effects of VH-N412 on learning and memory following SE**

Epilepsy is associated with learning and memory difficulties in patients and animal models (*Giovagnoli and Avanzini, 1999*; *Löscher and Stafstrom, 2023*). We thus assessed whether VH-N412-induced reduction of neurodegeneration and inflammation ameliorates mnesic capacities of our epileptic animals. Three weeks after SE, a group of mice was submitted to behavioral tests (*Figure 8*). We first evaluated hippocampus-dependent learning and memory performance using the olfactory tubing maze (OTM), a test particularly well suited for the FVB/N mouse strain we used, with poor or deficient vision (*Girard et al., 2016*). Using the OTM test, the intertrial interval (ITI) showed no significant effect across the five training sessions (multivariate analysis of variance [MANOVA]: $F_{(8,92)} = 1.16$, nonsignificant (ns) and between the three groups (MANOVA: $F_{(2,23)} = 1.85$; ns) (*Figure 8A*). However, considering the percentage of correct responses (MANOVA: $F_{(8,92)} = 0.99$; ns), a subsequent group difference was observed between the three groups (MANOVA: $F_{(2,23)} = 6.23$; p<0.01) (*Figure 8B*). Post hoc analysis using the Newman-Keuls test revealed that SE mice reached a significantly lower percentage of correct responses in comparison with the two other groups (p<0.05). Selective ANOVAs showed a significant difference starting from the third training session between SE and SE+VH-N412 groups (ANOVAs: $F_{(1,14)} \geq 5.12$; p<0.05). In addition, a similar difference was also observed between

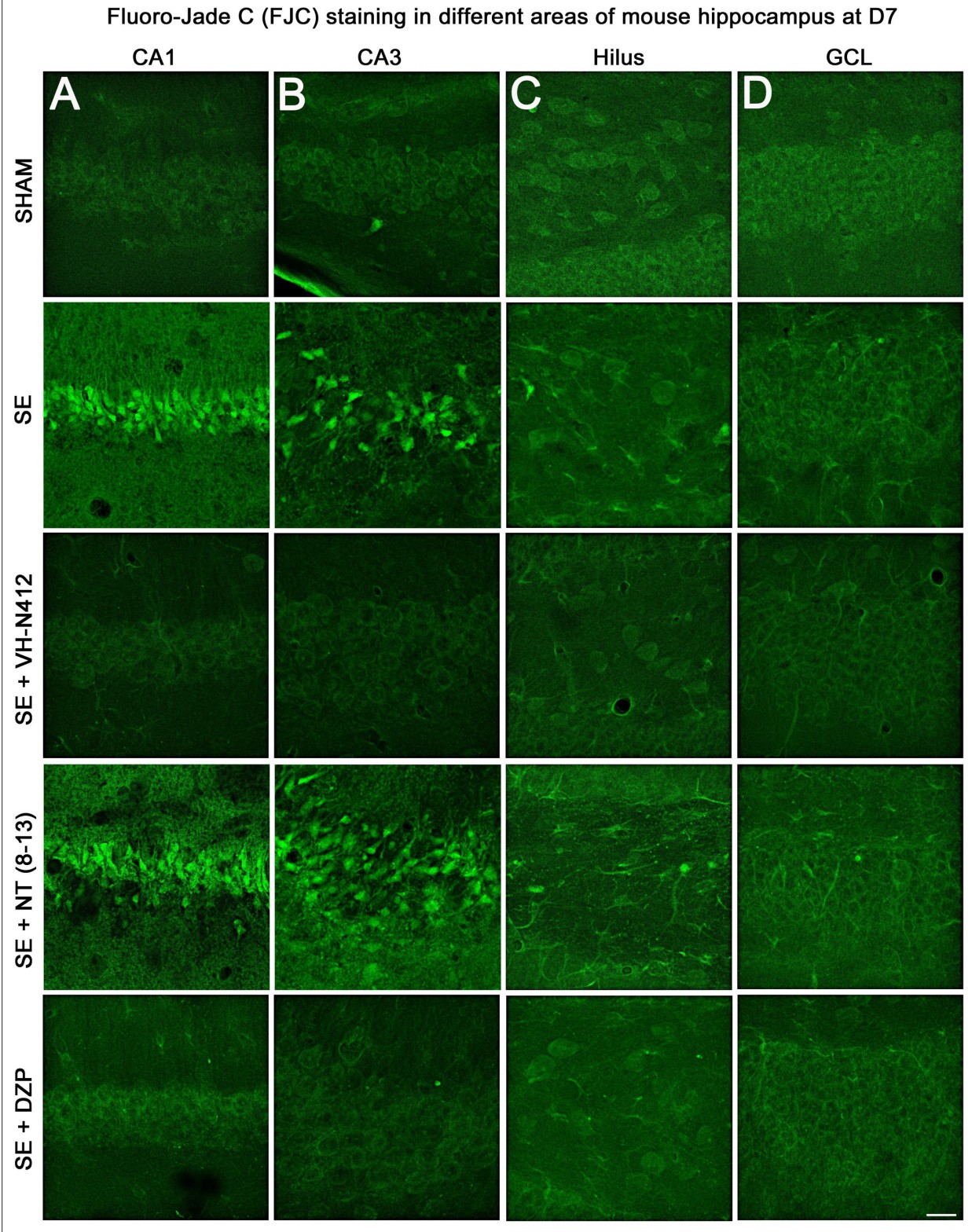

**Figure 5.** Representative examples of FJC staining in the different areas of mouse dorsal hippocampus. (**A**) CA1, (**B**) CA3, (**C**) H **and (D)** GCL, at D7 post-SE from SHAM, SE, SE+VH-N412, SE+ NT(8–13), and SE+DZP animals. Scale bar: 20 μm in all panels.

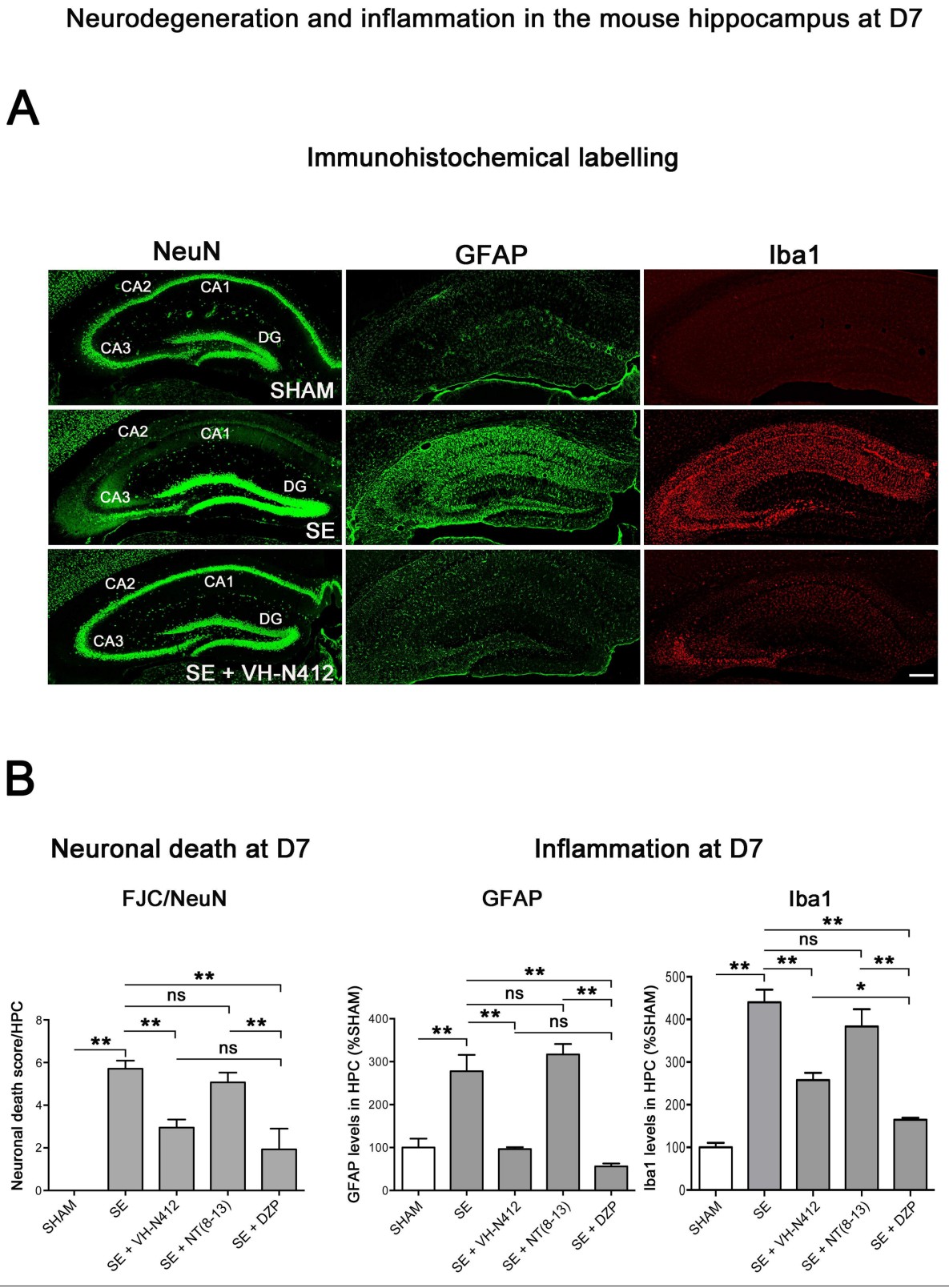

**Figure 6.** Neuroprotective and anti-inflammatory effects of VH-N412 following SE. (**A**) Immunohistochemical labeling was used to assess the extent of brain damage in coronal sections of the dorsal hippocampus from SHAM, SE, SE+VH-N412, SE+NT(8–13), and SE+DZP animals at D7 post-SE. Left panels show neurons labeled with the anti-NeuN antibody directed against a neuronal-specific nuclear protein in all animals. Middle and right panels show inflammation assessed with anti-GFAP and Iba1 antibodies to monitor astrocytic and microglial reactivity respectively. Scale bar: 200 µm in all

*Figure 6 continued on next page*

panels. In SHAM animals, a basal labeling for GFAP and Iba1 was detected in the hippocampus. In SE animals, a strong activation of glial cells occurred in all hippocampal layers. This inflammatory response was nearly abolished when VH-N412 was administered 30 min after SE onset. (**B**) Histograms comparing the mean neuronal death score, the mean GFAP, and Iba 1 levels in the dorsal hippocampus of SHAM, SE, SE+VH-N412, SE+NT(8–13), and SE+DZP animals. NeuN and FJC labeling were used to quantify neuronal death and the effects of VH-N412 (left histogram). The neuronal death score was expressed as the mean scores ± SEM. GFAP and Iba1 labeling levels allowed quantification of glial inflammation, which was expressed as the average percentage ± SEM normalized to CTL SHAM. In SHAM animals, no neuronal death was observed in the hippocampus (score 0). In SE animals, significant neuronal death was observed in CA1–3 pyramidal cell layers and H. Neurodegeneration observed in SE animals was significantly decreased when VH-N412 or DZP were administered 30 min after SE onset, but no changes were observed when NT(8–13) was administered. Asterisks indicate statistically significant differences: *p<0.05, **p<0.01 (Tukey's test).

SHAM and SE mice on session 5 (ANOVA: $F_{(1,14)}$ = 8.1; p<0.05) while no significant differences were observed between control and SE+VH-N412 mice (ANOVAs: $F_{(1,18)}$ ≤ 2.5; ns) during each of the five sessions.

Exploratory and spontaneous locomotor activity of the same mice was assessed in the open field paradigm (*Figure 8C*). SE mice exhibited a strong hyperactivity in comparison with SHAM mice. The average distances covered by mice from these two groups were significantly different on the first session of 5 min (ANOVA: $F_{(1,14)}$ = 29.56; p<0.001) and on the total of the two sessions (ANOVA: $F_{(1,14)}$ = 11.18; p<0.01). Treatment of SE mice with VH-N412 maintained a locomotor activity similar to SHAM mice on the two successive sessions (ANOVAs: $F_{(1,18)}$ ≤ 0.24; ns) and significantly lower than untreated SE mice on each of the two sessions of 5 min (ANOVAs: $F_{(1,14)}$ ≥ 5.92; p<0.05) (*Figure 8C*). No significant difference was observed between groups in the time spent in the center of the maze (data not shown). In all, we showed that VH-N412 treatment following SE preserved learning and memory capacities and normal locomotor activity in mice.

## VH-N412 does not modulate neuronal hyperactivity induced by KA in hippocampal slices

The effects of VH-N412 on seizure activity led us to question whether the conjugate could modulate hippocampal neuronal hyperactivity induced by KA. We addressed this question using acute hippocampal slices that were continuously perfused with ACSF preheated at 37°C. KA (300 nM) rapidly increased in the CA1 region, the spontaneous firing rate that remained rather steady over the 110 min of KA exposure. The normalized firing rate was 0.86±0.17 at the end of the experiment. When VH-N412 was applied at increasing doses of 0.1, 1, and 10 µM over a 30 min period, the KA-induced increase in firing rate in CA1 did not change significantly. Thus, the normalized firing rate was 0.96±0.11 after 20 min exposure to 0.1 µM VH-N412, 0.92±0.11 after 20 min exposure to 1 µM VH-N412, and 0.82±0.13 after 20 min exposure to 10 µM VH-N412. A slight and transient increase of the firing rate was observed just after exposure to 0.1 µM VH-N412 for two out of the four recorded slices (*Figure 8—figure supplement 1*).

## Expression and regulation of cold shock RBM3 and CIRBP mRNA and proteins

Cold shock proteins such as the RNA-binding protein RBM3 and cold-inducible RNA-binding protein (CIRBP) are involved in diverse physiological and pathological processes, including circadian rhythm, inflammation, neural plasticity, stem cell properties, and cancer development (reviewed in *Zhu et al., 2016*). In particular, cooling and hibernation in animals induces expression of RBM3 and CIRBP (*Shiina and Shimizu, 2020*) and boosting endogenous RBM3 levels through hypothermia is neuroprotective (*Ávila-Gómez et al., 2020*). We hence questioned whether the hypothermic and neuroprotective effects of VH-N412 were associated with RBM3 and CIRBP regulation in brain. A group of nine mice was administered VH-N412 at the dose of 4 mg/kg eq. NT leading to transient hypothermia. Mice were divided into three groups that were sacrificed at 4, 8, and 16 hr post VH-N412 administration. Brains were rapidly extracted, cut into two for mRNA and protein analysis. In each hemisection, the hippocampus was isolated and snap-frozen. RT-qPCR analysis normalized with GAPDH on pooled samples (three animals per time point) showed significant increase of mRNA encoding RBM3 (2.86±0.33, 3-fold) and CIRBP (1.53±0.56, 1.5-fold) relative to PBS injected control (1.00±0.00) at the 16 and 8 hr time points, respectively (Dunnett's test, p<0.05) (*Figure 9A*). Next, using western

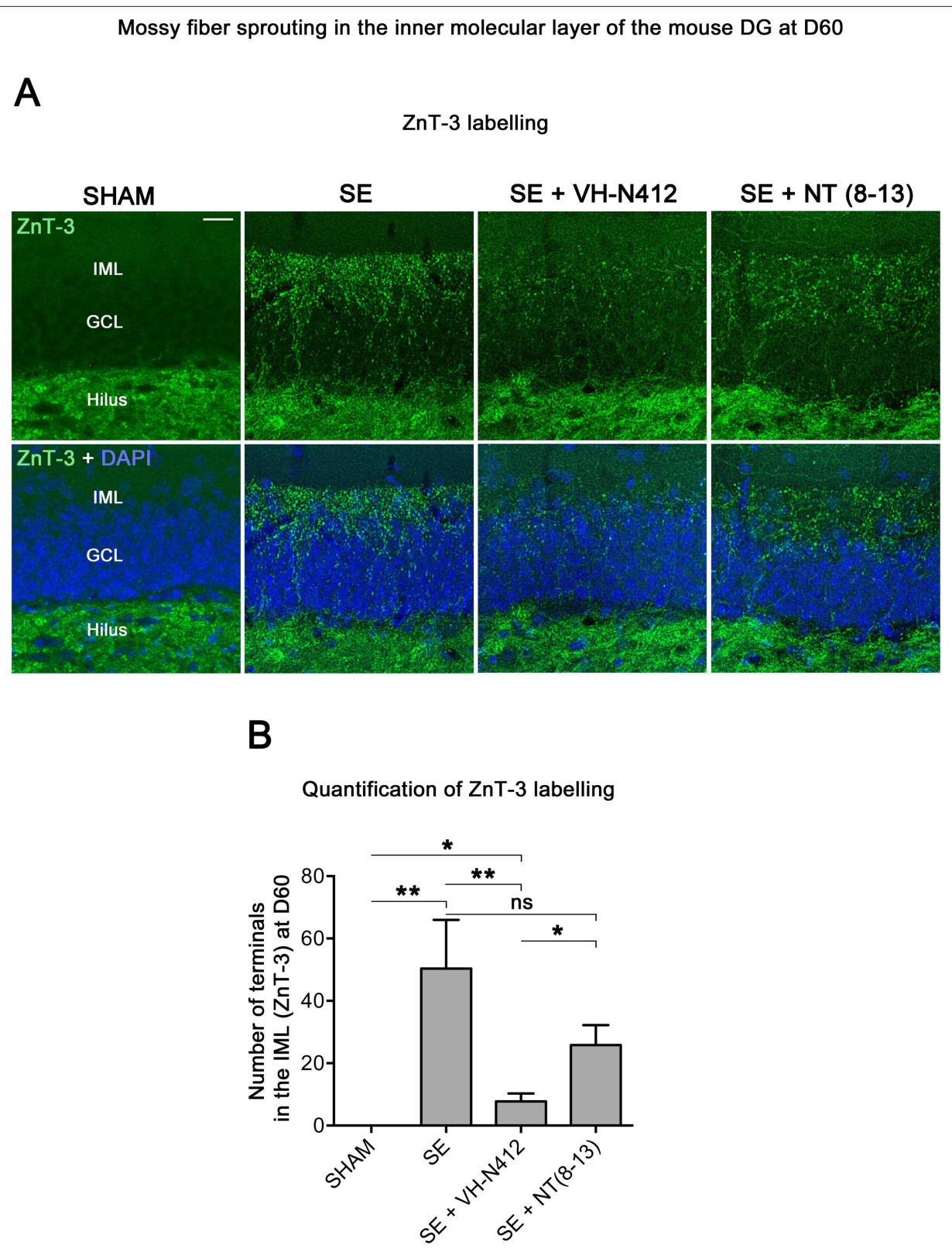

**Figure 7.** VH-N412 reduces mossy fiber sprouting in the hippocampus 8 weeks (D60) after SE. (**A**) The effects of VH-N412 on mossy fiber sprouting was assessed 8 weeks after induction of SE with immunohistochemical labeling for the zinc vesicular transporter 3 (ZnT-3). In SHAM animals, mossy fiber terminals were only present in the H. In SE animals, in addition to ZnT-3 staining present in the H, mossy fiber terminals were also observed within the IML. Scale bar: 20 μm in all panels. (**B**) Semiquantitative analysis revealed that ZnT-3 staining in the IML at D60 was significantly reduced in animals

*Figure 7 continued on next page*

*Figure 7 continued*
administered with VH-N412 30 min after SE onset, but unchanged when NT(8–13) was administered. Asterisks indicate statistically significant differences: *p<0.05, **p<0.01 (Tukey's test).

blot and antibodies that detect RBM3, CIRBP, and β-actin used as an internal control, we evaluated in hippocampal samples whether increased mRNA levels translate into increased protein levels. Results showed significant increase of RBM3 protein levels (4 hr: 270.88 ± 5.12%, 2.7-fold, Dunnett's test, p<0.05; 8 hr: 465.74 ± 70.05%, 4.65-fold Dunnett's test, p<0.001; 16 hr: 251.97 ± 49.34%, 2.5-fold, Dunnett's test, p<0.05) and CIRBP protein levels (4 hr: 235.15 ± 13.02%, 2.35-fold, Dunnett's test, p<0.001) relative to PBS injected control (100.00 ± 0.00%) (*Figure 9B*, *Figure 9—source data 1*). However, CIRBP protein levels were not significantly altered at the 8 hr (157.24 ± 24.55%) and 16 hr (109.79 ± 10.15%) time points relative to PBS injected control (100.00±0.00, p>0.05; ANOVA) (*Figure 9B*, *Figure 9—source data 1*).

To confirm that it is indeed hypothermia rather than VH-N412 that contributed to modulation of RBM3 and CIRBP mRNA levels in vivo, we incubated cultured primary rat hippocampal neurons with three concentrations of VH-N412 (0.1, 1, and 10 µM) for 24 hr and assessed steady-state levels of RBM3 and CIRBP mRNA by RT-qPCR. Our results showed that cultured neurons treated with VH-N412 did not display significant differences of RBM3 and CIRBP mRNA levels at all concentrations compared to nontreated cultures (p>0.05; ANOVA, *Figure 9C*).

## Hippocampal expression of NTSR1 in vitro and in vivo

One important question raised by our results is whether NTSR1 receptors were indeed expressed in hippocampal neurons on which VH-N412 could exert some of its effects. To address this question, we performed immunocytochemistry and immunohistochemistry using a commercially available goat NTSR1 polyclonal antibody. We first validated the specificity of this antibody by using transfection experiments followed by immunocytochemistry in different cell types from different species, notably in human HEK 293 cells and rat hippocampal neurons cultured 21 days in vitro (21 DIV) (*Figure 10*).

Both HEK 293 cells (*Figure 10A and B*) and hippocampal neurons (*Figure 10C and D*) displayed higher NTSR1 immunolabeling (green) after transfection with a plasmid construct encoding rat NTSR1 compared to non-transfected cells (see arrows, cells labeled for DAPI but not for NTSR1). In addition, both types of cells exhibited high NTSR1 immunostaining within the cell body with a punctate pattern (c2 high-magnification inset) and at the plasma membrane (arrows in E, F, c1, f1 high-magnification insets), as expected for receptor localization. The axons, the dendritic arbors, and protuberances of hippocampal neurons were also immunostained (*Figure 10C and D*, arrowheads in f2 high-magnification inset), in agreement with *Boudin et al., 1998*; *Pickel et al., 2001*. In the boxed areas, NTSR1 labeling (f3a,3b high-magnification insets) and RFP (red), used to underline dendritic structures (f4a, 4b high-magnification insets), confirmed that NTSR1 was also located in dendritic spines since NTSR1 and RFP proteins colocalized (f5a, 5b high-magnification insets). However, some of the NTSR1 labeling was slightly shifted relative to RFP, suggesting its localization at the cell membrane (stars in f3a-f5b high-magnification insets).

### Expression of endogenous NTSR1 in cultured hippocampal neurons

To assess the expression of endogenous NTSR1 and its localization, 21 DIV cultured hippocampal neurons were fixed and immunostained sequentially with the goat anti-NTSR1 polyclonal (red), the mouse anti-MAP2 antibody (blue), and a rabbit anti-drebrin E/A antibody (green) (*Figure 11*). Mature cultured hippocampal neurons displayed high levels of endogenous NTSR1 with a punctate pattern (*Figure 11B and K*, red) at high magnification of the boxed area in *Figure 11A and F*. Enlargement of the boxed area in *Figure 11J and L* illustrated that NTSR1 was closely apposed to dendritic shafts and spines, presumably at the level of the cell membrane, as revealed by the neuronal and dendritic shaft marker MAP2 (*Figure 11E and H*, blue), and dendritic spine marker drebrin (*Figure 11I, H, J, and L*, green, arrows in high-magnification insets). Note that no NTSR1 immunostaining was observed in filopodia (small arrow in *Figure 11H, J, and L*).

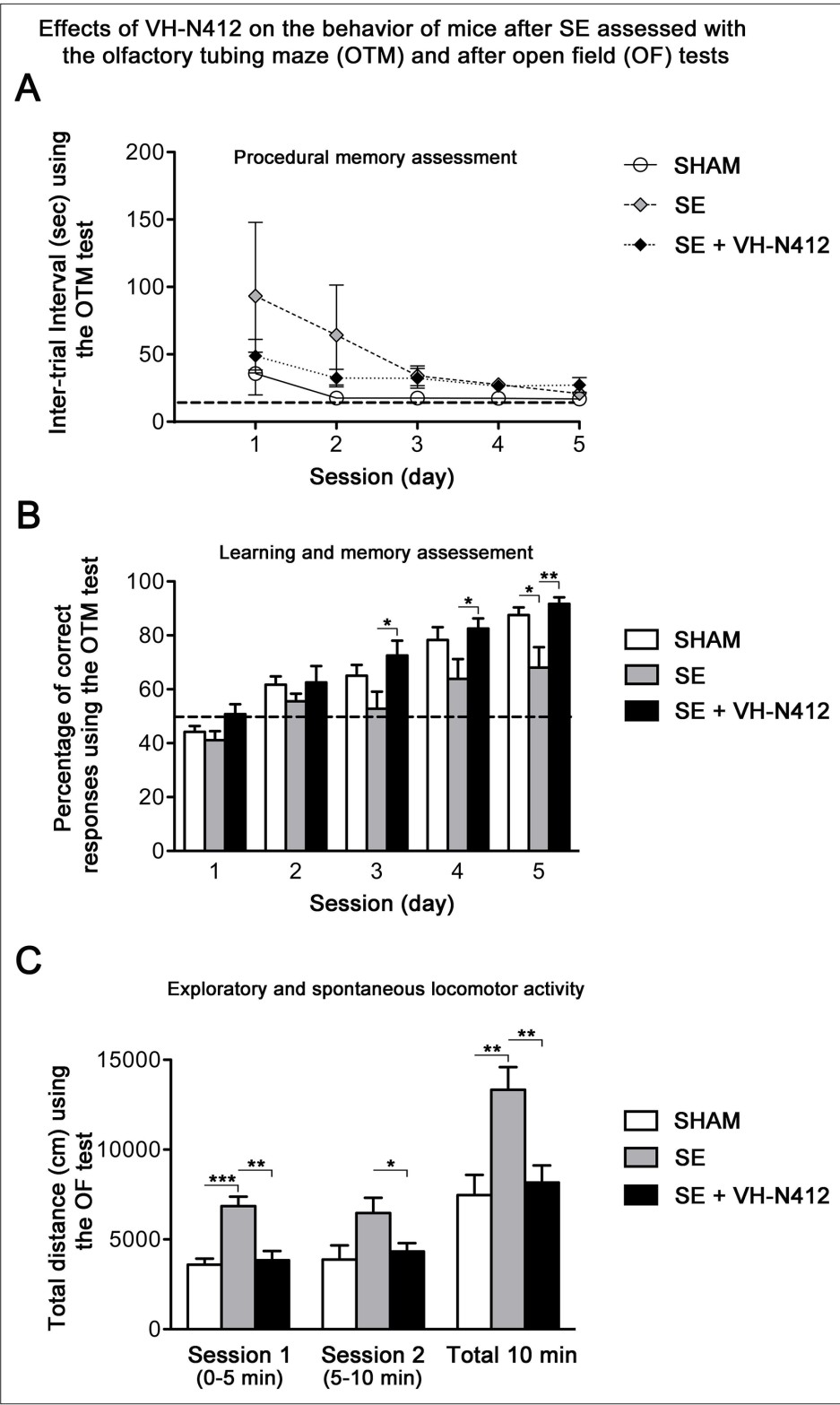

**Figure 8.** Effects of VH-N412 on the regulation of RBM3 and CIRBP mRNAs/proteins in the hippocampus and in vitro. (**A**) Histograms related to analyses of RBM3 and CIRBP mRNA levels in the hippocampal tissue of CTL and mice treated with VH-N412 at 4, 8, and 16 hr by RT-qPCR. Note that VH-N412 treatment increased the expression of RBM3 at 16 hr while CIRBP peaked at 8 hr compared to CTL mice. (**B**) Western blots of CIRBP and RBM3 protein expression in VH-N412-treated mice at the same time points as above. Note that CIRBP and RBM3 were

*Figure 8 continued on next page*

*Figure 8 continued*

expressed predominantly at their expected molecular weight (MW) of ~18 kDa (stars). However, bands of higher MW between ~22 kDa (RBM3) and~25 kDa (CIRBP) were also detected in agreement with ***Rosenthal et al., 2017***; ***Zhu et al., 2024***. Similar patterns of expression of the higher MW bands compared to the ~18 kDa band strongly suggest that they correspond to either different isoforms for each protein or result from post-translational modifications. Protein levels were normalized with β-actin (~42 kDa, star) in all samples, and samples from mice treated with VH-N412 at 4 (n=3; lanes 4–6), 8 (n=3; lanes 7–9), and 16 hr (n=3; lanes 10–12) were normalized with CTL mice treated with PBS (CTL PBS; n=3; lanes 1–3). The graphs correspond to quantification of the western blots. RBM3 is upregulated at 4, 8, and 16 hr in VH-N412-treated mice, whereas CIRBP is upregulated at 4 hr after treatment. These results indicated that in vivo, VH-N412-induced hypothermia was associated with a regulation of cold stress proteins. (**C**) In vitro quantification of mRNA levels of RBM3 and CIRBP analyzed by RT-qPCR after treatment with different concentrations of VH-N412 (0.1, 1, and 10 μM) of cultured hippocampal neurons. VH-N412 did not regulate RBM3 and CIRBP mRNA at all concentrations used in our cultured hippocampal neurons.

The online version of this article includes the following figure supplement(s) for figure 8:

**Figure supplement 1.** MEA electrophysiological recordings of acute hippocampal slices stimulated with KA in the presence or absence of VH-N412.

## NTSR1 is expressed in vivo, in adult mouse hippocampus

To confirm that the NTSR1 protein was expressed in the hippocampus in vivo, immunohistochemical labeling for NTSR1 was performed on control coronal sections of adult mice (3–5 months, *Figure 12*). At low magnification, NTSR1 immunolabeling was homogeneous in all areas and layers of the hippocampus (*Figure 12A*, DAPI in blue) and displayed a regional- and laminar-specific pattern within the hippocampus (*Figure 12B*, green). Moderate to strong NTSR1 immunolabeling was observed in all layers, including the stratum oriens (O), stratum radiatum (R), and the stratum lacunosum-moleculare (LM) of the CA1-CA2-CA3 areas, the stratum lucidum (SL) of CA2-CA3, the molecular layer (M) and H of the DG. The hippocampal pyramidal neurons (P) of CA1, CA2, and CA3 were relatively more immunostained as compared to GCL of the DG (*Figure 12B*). At high magnification, NTSR1 immunolabeling displayed a punctate pattern in CA1 (*Figure 12C*). NTSR1 was expressed at the plasma membrane of CA1 cell bodies (arrowheads in *Figure 12D and F*), consistent with the cell membrane localization of NTSR1 as well as within proximal dendrites of pyramidal neurons (arrows in *Figure 12D and F*). Note that similar to transfection results and immunocytochemistry on mature cultured neurons, the dendritic protuberances, reminiscent of dendritic spines, were also highly immunolabeled for NTSR1 (arrowheads in *Figure 12D* high-magnification inset).

## Effects of VH-N412 on hippocampal neuronal survival and on dendrite length following NMDA or KA intoxication

Since SE-induced neurodegeneration was significantly decreased by VH-N412 (*Figure 6A and B*), we investigated in vitro whether these effects were associated with intrinsic neuroprotective effects of the conjugate. For this purpose, we induced intoxication of cultured hippocampal neurons by NMDA or KA treatment (***Mattson et al., 1995***; ***Kajta and Lasoń, 2000***) and evaluated whether VH-N412 elicited neuroprotection by assessing hippocampal neuronal survival (left histogram) and total dendrite length (right histogram). Note that the NTSR antagonists and VH-N412 alone showed no significant toxic effects on hippocampal neuronal survival and total dendrite length at all concentrations tested (*Figure 13A*). NMDA (100 μM, 10 min) induced significant neuronal death (51.77% of the CTL, p<0.001, Dunnett's test, *Figure 13B*). VH-N412 at 1 and 10 μM protects significantly hippocampal neurons from NMDA-induced cell death compared to control (75.66% and 70.28% of the CTL, respectively, p<0.05, Dunnett's test). These effects of VH-N412 were as potent as those of BDNF (50 ng/mL) (72.28% of the CTL, p<0.05, Dunnett's test), used as a well-known neuroprotective molecule following NMDA intoxication (***Mattson et al., 1995***). In contrast to BDNF, we did not observe significant effect of VH-N412 on total dendrite length following NMDA intoxication (data not shown). Antagonizing NTSR with the SR142948A (0.1; 1 and 10 μM) and SR48692 (0.1 and 1 μM) NTSR antagonists blocked the neuroprotective effects mediated by VH-N412 in rat primary neuronal cultures injured by NMDA (*Figure 13B*).

We next analyzed the effects of VH-N412 on hippocampal neuronal survival (*Figure 13C*, left histogram) and on total dendrite length (*Figure 13C*, right panel) following another neuronal intoxication

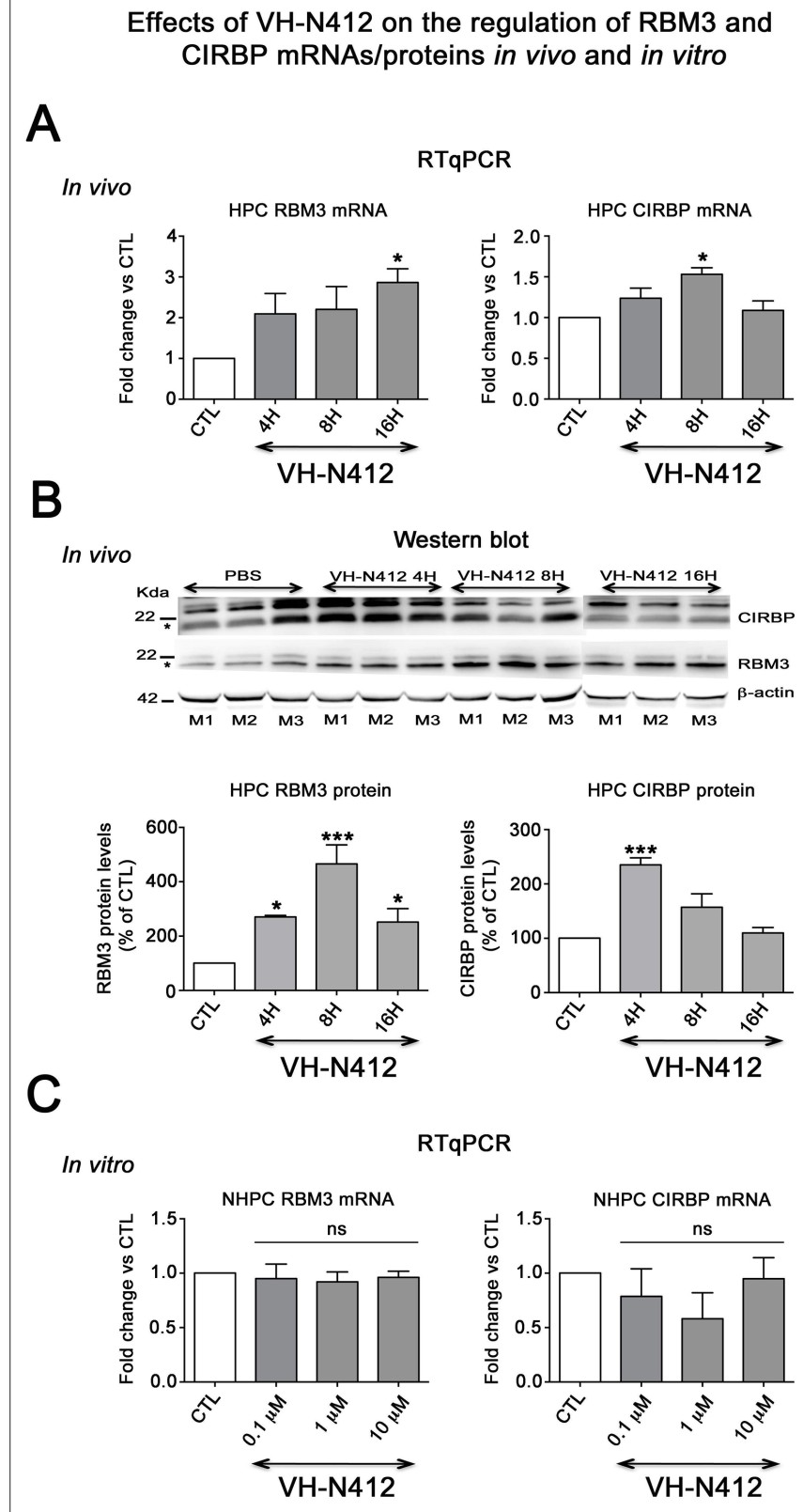

**Figure 9.** Following SE, VH-N412 preserved hippocampus-dependent learning and memory and normal locomotor activity. (**A and B**) Learning and memory performance was assessed from 3 to 4 weeks after SE using the olfactory tubing maze (OTM). (**A**) Illustrates the mean ITI between the 12 trials in the OTM (in seconds, ± SEM). The dashed line indicates the minimum fixed ITI (15 s). There was no difference between groups on the ITI. (**B**) Mean

*Figure 9 continued on next page*

*Figure 9 continued*

percentage of correct responses obtained in the OTM during five training sessions of 12 trials per day. The dashed line denotes the chance level (%). From the second session, all animal groups had learned and memorized the test tasks. Only after the fifth session did the epileptic SE mice (n=6) show a significant impairment in memorization and learning while SE mice treated with VH-N412 showed similar performance to that of SHAM mice (n=10). (C) Locomotor activity using the open field (OF) test. (C) Illustrates the mean traveled distance in centimeters during two consecutive 5 min sessions (sessions 1 and 2) using OF test. SE mice (n=6) displayed a strong and significant hyperactivity in comparison with SHAM (n=10) while SE+VH-N412 mice (n=10) exhibited significant reduced exploratory and spontaneous locomotor activity, similar to that observed in SHAM mice.

The online version of this article includes the following source data for figure 9:

**Source data 1.** Uncropped immunoblot exposures displayed in *Figure 9B*.

model based on KA. As observed in *Figure 13C*, KA (25 µM, 24 hr) induced significant neuronal death (66.88% of the CTL, p<0.001, Dunnett's test). VH-N412 at 0.1, 1, and 10 µM significantly protected hippocampal neurons from KA-induced cell death compared to control (80.46%, p<0.05, 85.88%, p<0.01, 91.23%, p<0.01 of the CTL, respectively, Dunnett's test). These effects of VH-N412 were similar to those of oestradiol (100 nM) (81.68% of the CTL, p<0.05, Dunnett's test), used as a neuroprotective molecule following KA intoxication (*Kajta and Lasoń, 2000*). In contrast to oestradiol, VH-N412 had no significant effects on the total length of dendrites following KA injury. Altogether, our data suggested that VH-N412 elicited neuroprotective effects by modulating hypothermia, as observed in vivo, but also displayed intrinsic neuroprotective properties that were temperature independent.

## Discussion

Research conducted in preclinical and clinical settings demonstrated that mild to moderate hypothermia can provide neuroprotection in situations where there is an increased risk of neuronal death. These situations include sudden cardiac arrest followed by resuscitation, ischemic stroke, perinatal hypoxia/ischemia, TBI (*Kida et al., 2013*; *Andresen et al., 2015*), and seizures (*Sartorius and Berger, 1998*; *Schmitt et al., 2006*; *Niquet et al., 2015a*; *Niquet et al., 2015b*) in animal models and in humans (*Karkar et al., 2002*). Conversely, hyperthermia aggravates SE-induced epileptogenesis and neuronal loss in immature rats (*Suchomelova et al., 2015*). Hypothermia may open new therapeutic avenues for the treatment of epilepsy and for the prevention of its long-term consequences.

Intracerebral administration of NT or NT(8–13) analogues was associated with PIH (*Bissette et al., 1976*; *Coquerel et al., 1988*; *Coquerel et al., 1986*; *Fanelli et al., 2015*). When NT is administered peripherally, it is quickly metabolized by peptidases and has limited access through the BBB (*McMahon et al., 2002*; *Gordon et al., 2003*; *Orwig et al., 2009*; *Boules et al., 2013*). The conjugation of vector molecules can enhance the transport of active components across the BBB (*Pardridge, 2001*; *Pardridge, 2003*; *de Boer and Gaillard, 2007*; *Jones and Shusta, 2007*; *Pardridge, 2007*; reviewed in *Vlieghe and Khrestchatisky, 2013*). Our objective was to generate 'vectorized' forms of NT that crossed the BBB and that displayed potent hypothermic properties, and to assess their potency in experimental epileptic conditions. For this purpose, we generated several conjugates that encompass peptides that target the LDLR and short active variants of NT. These were compared for their potential to bind LDLR on the one side and NTSR1 on the other using biophysical, cellular in vitro and in vivo approaches, and on their potential to induce hypothermia in different species, including mice, rats, and pigs. Based on this comparison, we selected the VH-N412 conjugate for further studies and evaluated its neuroprotective properties in a mouse model of temporal lobe epilepsy. We showed that this compound reduced epileptic seizures, neurodegeneration, neuroinflammation, mossy fiber sprouting, and preserved learning and memory skills in the epileptic mice. We showed that the target receptor NTSR1 was expressed in hippocampal pyramidal neurons in vitro and in vivo, in cell bodies, dendrites, and spines. Besides the neuroprotective hypothermia effects induced in vivo with VH-N412, we showed in cultured hippocampal neurons that this compound also displayed temperature-independent neuroprotective properties.

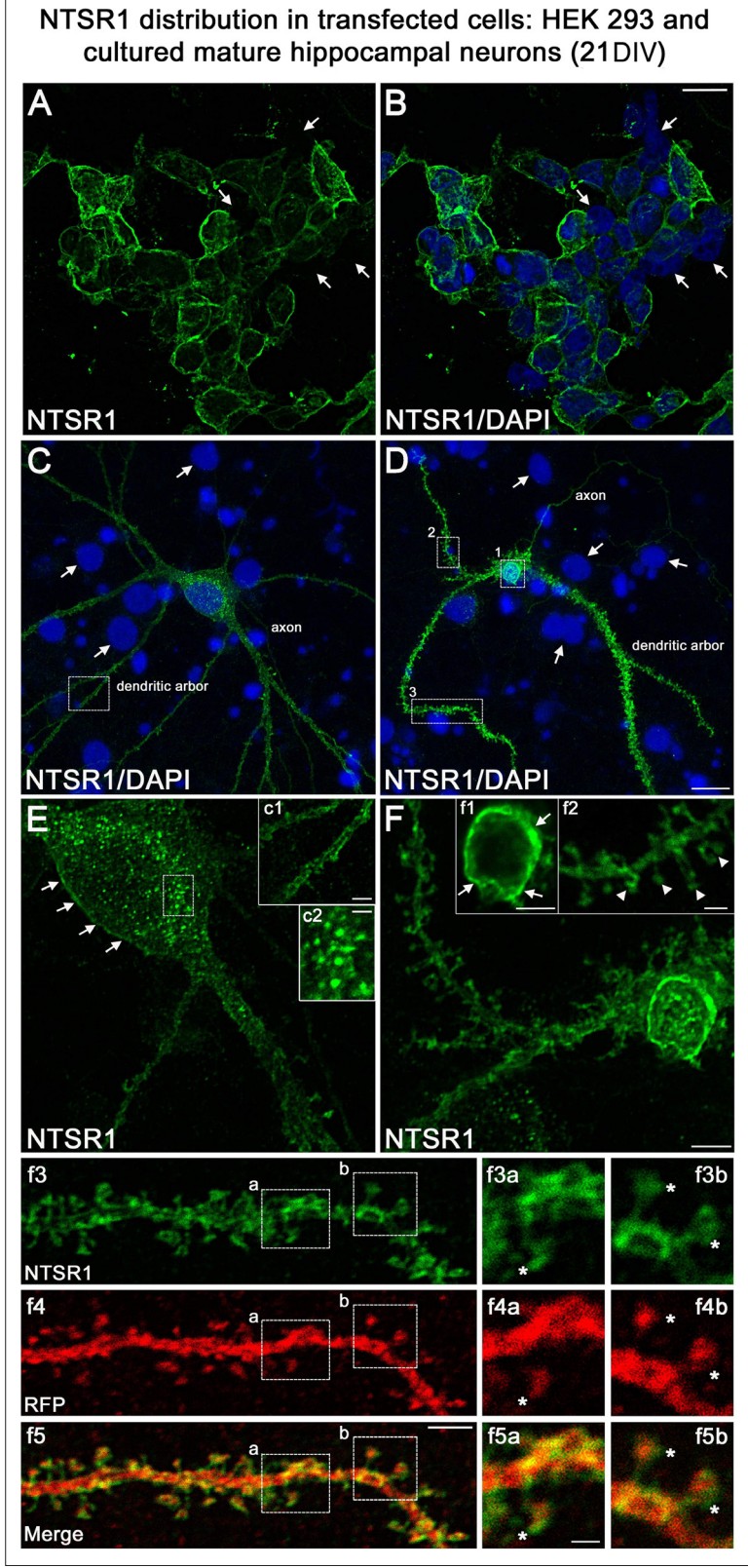

**Figure 10.** Validation of NTSR1 antibody. The specificity of the goat polyclonal NTSR1 antibody was assessed by using transfection experiments for 43 hr followed by immunocytochemistry in different cell types including (**A and B**) human HEK 293 cells and (**C and D**) rat cultured mature hippocampal neurons (21 DIV). Cell nuclei were labeled with DAPI (blue). Both HEK 293 and hippocampal neurons displayed stronger NTSR1 immunolabeling

*Figure 10 continued on next page*

*Figure 10 continued*

(green) after transfection with a plasmid construct encoding rat NTSR1 (see cells double labeled for NTSR1 and DAPI), compared to non-transfected cells (see arrows, cells labeled for DAPI but not for NTSR1). Moreover, both types of cells exhibited high NTSR1 immunostaining within the cell body with a punctate pattern (see c2) and at the plasma membrane (see arrows in E, F, c1, and f1) as expected for receptor localization. The axons, the dendritic arbors, and their protuberances (see arrowheads in f2) of hippocampal neurons were also immunostained. f3 and f4 correspond to the dendritic portion of a neuron overexpressing NTSR1 (green) and RFP (red). RFP was used to outline the morphology of neurons including the dendrites and their dendritic spines. f5 corresponds to the merge of panels f3 and f4. Panels f3a and f3b correspond to the high magnification of NTSR1 labeling in two distinct areas of a dendrite (boxed in f3, f4, f5). Panels f4a and f4b correspond to RFP labeling in these same areas. Panel f5a corresponds to the merge of f3a and f4a. Panel f5b corresponds to the merge of f3b and f4b. Double immunostaining of NTSR1/RFP confirmed that NTSR1 was located in dendritic spines. However, some of the NTSR1 immunolabeling was slightly shifted relative to RFP (see stars in panels f3a to f5b) suggesting NTSR1 localization in the cell membrane. Scale bars: 20 µm in A and B; 5 µm in C, D, E, F, c1, f1–f5; 2 µm in f3a–f5b.

## LDLR-binding peptides conjugated to NT induced potent hypothermia in naïve mice when administered systemically

We coupled LDLR-binding peptides to NT and optimization and SAR evaluations were performed to facilitate synthesis together with improving biological properties of the conjugate, namely LDLR and NTR1 binding, metabolic stability, BBB-crossing and, in the end, central hypothermic potential after systemic administration in mice. Starting with the VH-N21 and VH-N41 conjugates encompassing respectively the VH445 and VH4129 peptides conjugated to the full-length NT tridecapeptide, we selected the VH-N412 conjugate encompassing the VH4129 and NT(8–13) peptides, to generate a low molecular weight conjugate (1929kDa) that displayed the most potent and sustained hypothermic potential in naïve mice at low dose (ED50 ca. 0.8mg/kg, corresponding to 0.7mg/kg eq. NT). Consistent with previous reports with other LDLR-binding VH445 peptide analogues (*Jacquot et al., 2016*; *David et al., 2018*; *Varini et al., 2019*), the VH4129 peptide retained its binding potential to LDLR when associated to the NT(8–13) peptide (63.8nM), compared to the free VH4129 (64.8nM). On the other hand, binding of the NT(8–13) fragment to the NTSR-1, which was previously shown to mediate central hypothermia (*Pettibone et al., 2002*; *Remaury et al., 2002*; *Mechanic et al., 2009*), was also retained when associated to VH4129. Interestingly, the VH4129 peptide analogue was evaluated based on its optimal LDLR-binding profile, with moderate affinity (similar to the VH445 analogue) but with faster on-rate and off-rate than VH445 (*Jacquot et al., 2016*), endowing this analogue with a better binding/release profile for BBB-crossing. Accordingly, the BBB influx rate ($K_{in}$) of the VH-N412 compound was 3-fold higher than the initial VH-N21 compound encompassing the VH445 peptide, and 10-fold higher than the native NT peptide. Although only sparse data exists evaluating this parameter for NT analogues (*Banks et al., 1995*), this result confirmed the optimized brain uptake potential of the VH-N412 compound. Finally, because the primary cleavage site of the NT(8–13) peptide in plasma is at the N-terminal Arg-Arg site (*Orwig et al., 2009*), conjugation of the VH4129 peptide probably protected to some extent from aminopeptidases. This was confirmed by the extended metabolic resistance of VH-N412 in vitro in both mouse and human blood, with more than 1hr half-life vs 44min with the VH445, previously found to be less stable than the optimized VH4129 analogue (*Jacquot et al., 2016*), and only 9min for the native NT. This was also consistent with the rapid enzymatic proteolysis of endogenous peptides and generally small linear peptides containing only natural amino acids, highlighting the advantage of using a fully optimized LDLR-binding peptide (*Foltz et al., 2010*). The in situ brain perfusion results showed clearly that the VH-N21, and more so the VH-N412, displayed higher brain uptake in mice than nonconjugated NT. However, we cannot exclude that the hypotherma we observed with the different conjugates was related in part to increased stability conferred to NT.

## VH-N412 induced potent PIH, and attenuated seizures, neurodegeneration, and neuroinflammation in the KA model of epilepsy

Following induction of SE with KA, we showed that the VH-N412 compound elicited rapid hypothermia that was associated with anticonvulsant effects. In particular, we observed potent neuroprotection,

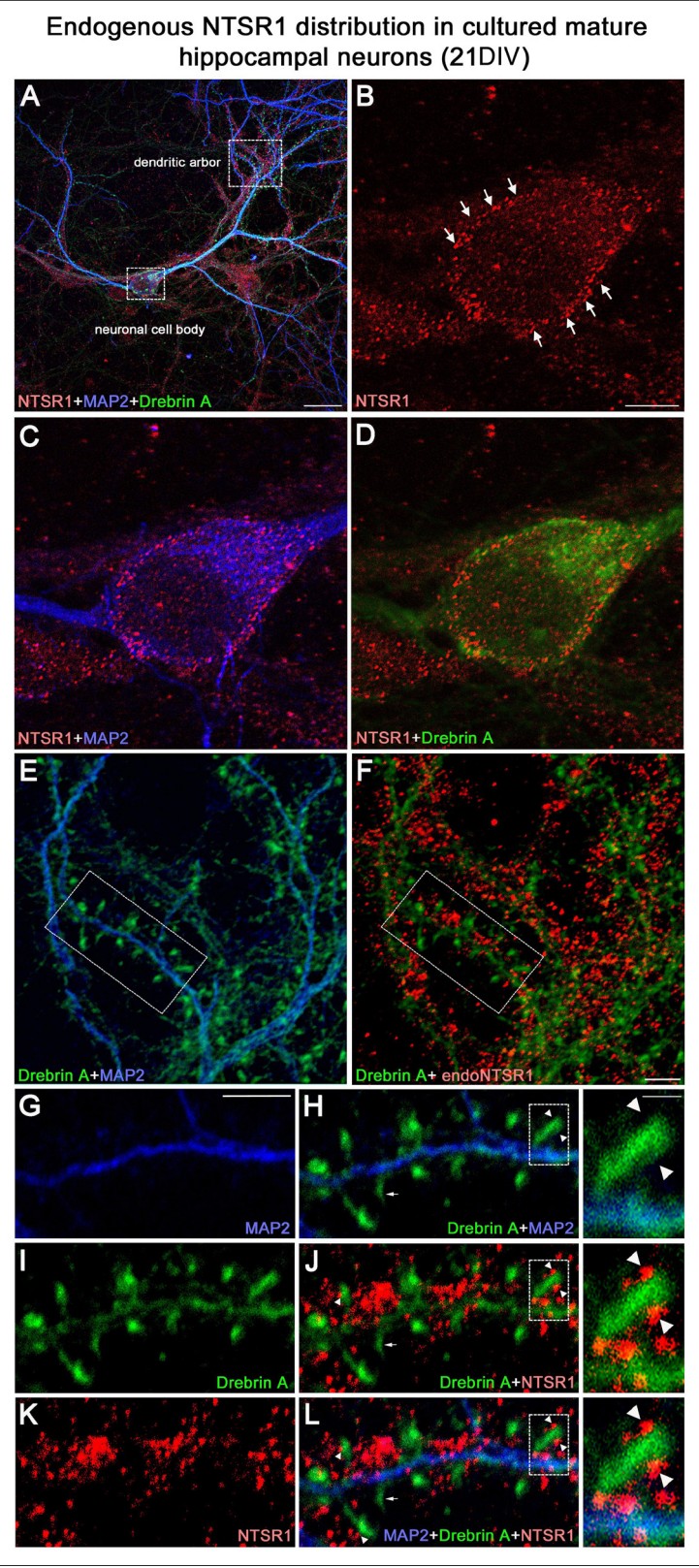

**Figure 11.** Expression of endogenous NTSR1 and its localization in mature cultures of hippocampal neurons. Twenty-one-day-old (21 DIV) cultured hippocampal neurons were fixed and immunostained sequentially with antibodies against NTSR1 (red), MAP2 (blue), and drebrin E/A (green). Panels **A, L** and high magnification of the boxed area in L correspond to the merge of NTSR1/MAP2/drebrin. (**E, H** and **F, J**) Merge of MAP2/drebrin and

*Figure 11 continued*

NTSR1/drebrin respectively. At high magnification of the boxed-in area in A (pyramidal neuron cell body) and F (dendrites), mature cultured hippocampal neurons displayed endogenous NTSR1 with a punctate pattern (**B, C, D, F, J, K, L**, red) similar to observations in transfected cells (*Figure 10*). Enlargement of the boxed-in area in F and J illustrated that NTSR1 (red) was closely apposed to dendritic shafts and dendritic spines, presumably at the level of the cell membrane, as revealed by neuronal and dendritic shaft marker MAP2 (E and H, blue) and dendritic spine marker drebrin (**I, J,** and **L**, green, see arrows in high-magnification insets). Note that no NTSR1 immunostaining was observed in filopodia (see small white arrows in **H, J,** and **L**). Scale bars: 20 µm in A; 5 µm in B–L; 1 µm in boxed-in area in H, J, and L panels.

reduced inflammation in the hippocampus, reduced sprouting of the DG mossy fibers, and preserved learning and memory skills in the epileptic mice treated with VH-N412. To our knowledge, these results are the first to show a significant impact of PIH in an epileptic mouse model. Hypothermia appears to alleviate seizure intensity very rapidly, as early as 30min after VH-N412 administration, and as efficiently as DZP, one of the most efficient anti-seizure drugs that interestingly also induces hypo-thermia (*Irvine, 1966*). Several studies showed that NT or stabilized NT analogs displayed anticonvul-sant properties while hypothermia was apparently not evaluated in these studies (*Green et al., 2010*; *Lee et al., 2009*; *Robertson et al., 2011*; *Clynen et al., 2014*). We thus cannot exclude that to some extent, our NT-peptide conjugates also reduce seizures independently of hypothermia.

The neuroprotective effects we observed are likely due to reduced seizure burden but also possibly to reduced brain metabolism during the phases that follow seizures. Such neuroprotective effects are comparable to those discussed below in several studies of acute brain damage in rodent models of hypoxia-ischemia, TBI, and intracerebral hemorrhage (ICH) using different NT analogs. Among these, NT77 was modified from NT at amino acid positions located in the hexapeptide of NT[8–13] (*Boules et al., 2001*). NT77 crossed the BBB, induced hypothermia (*Gordon et al., 2003*), reduced oxidative stress in the rat hippocampus (*Katz et al., 2004a*), and improved neurological outcome after asphyxial cardiac arrest (*Katz et al., 2004b*). ABS-201, also known as HPI-201, another synthetic analogue of NT[8–13], has been described as a second-generation high-affinity NTSR1 agonist that exhibited BBB permeability, and effectively induced PIH in a number of experimental paradigms. In a focal ischemic model of adult mice ABS-201 induced recovery of sensory motor function (*Choi et al., 2012*). It also promoted the integrity of the BBB and the neurovascular unit (*Zhao et al., 2020*). HPI-201 greatly enhanced the efficiency and efficacy of conventional physical cooling in a rat model of ischemic stroke (*Lee et al., 2016b*). In the same model in mice and oxygen glucose deprivation in cortical neuronal cultures, HPI-201 displayed anti-inflammatory effects (*Lee et al., 2016a*). HPI-201-induced hypo-thermia saved neurons and endothelial cells inside the ischemic core in mice (*Jiang et al., 2017*). In a rat model of ventricular fibrillation cardiac arrest, ABS 201-induced TH, ameliorated post-resuscitation myocardial-neurological dysfunction, and prolonged survival duration (*Li et al., 2019*). HPI-201 was effective in reducing neuronal and BBB damage, attenuating inflammatory response and detrimental cellular signaling, and promoting functional recovery after TBI in the developing rat brain (*Gu et al., 2015*). HPI-201 also protected the brain from ICH injury in mice (*Wei et al., 2013*). Another second-generation NTSR1 agonist, HPI-363 elicited hypothermia and protective effects, improving sensorim-otor functional recovery in a TBI model in adult rat brain (*Lee et al., 2014*) and induced beneficial effects on chronically developed poststroke neuropsychological disorders, in the prefrontal cortex of mice (*Zhong et al., 2020*).

## NT receptors potentially involved in VH-N412-induced PIH in the KA model of epilepsy

NT-producing neurons and their projections are widely distributed in the brain, including the ante-rior hypothalamus (*Schroeder and Leinninger, 2018*) involved in thermoregulation. It is likely that NT-induced hypothermia acts via similar processes in the epilepsy model used in this study and in the different models of acute brain damage mentioned above, reducing both excitotoxicity and neuroin-flammation in different vulnerable regions of the CNS. While the mechanisms leading to hypothermia remain largely unknown, the NT analogues are known to modulate the activity of NT receptors. mRNA encoding the high-affinity NTSR1 was detected essentially in neurons in many hypothalamic regions, including the preoptic, anterior, periventricular, ventromedial, and arcuate nuclei (*Alexander and*

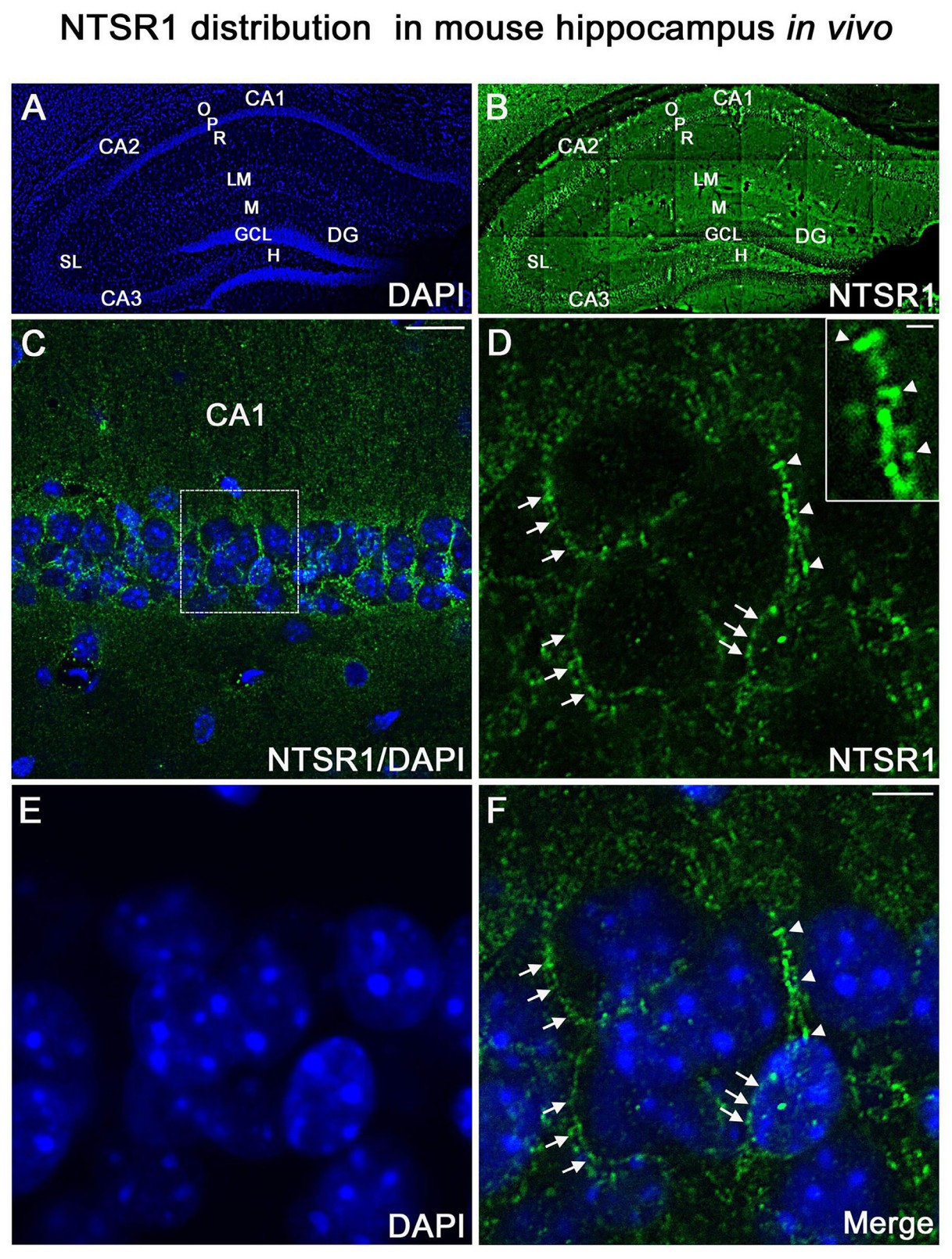

**Figure 12.** NTSR1 immunolabeling in mice hippocampal formation. (**A and B**) Correspond to low-magnification pictures showing coronal section of the mouse dorsal hippocampus processed with DAPI (blue), used to highlight the different cell layers of the hippocampus and NTSR1 (green) antibody respectively. Moderate to strong NTSR1 immunolabeling levels were found in the stratum (O), (R), (LM), (SL), in pyramidal neurons of CA1, CA2, and CA3, M, H, and GCL of the DG. (**C**) Merge of DAPI and NTSR1 (green) of the CA1 area at high magnification. (**D–F**) High magnification of the boxed-

*Figure 12 continued on next page*

*Figure 12 continued*

in CA1 area illustrating pyramidal neurons immunolabeled with NTSR1 antibody (**D**, green) and counterstained with DAPI (**E and F**, blue). NTSR1 immunoreactivity was observed in the cell bodies, at the cell membranes (see arrowheads in **E and F**), as well as at the proximal dendrites of CA1 pyramidal neurons (see arrows in **D and F**). NTSR1 immunolabeling displayed a punctate pattern. Note that several dendritic protuberances displayed high levels of NTSR1 immunolabeling (see arrows in inset in **D**). (**F**) Merge of NTSR1/DAPI. Scale bars: 225 µm in A and B; 20 µm in C; 5 µm in D-F; 1 µm in inset in D.

*Leeman, 1998*; *Nicot et al., 1994*; *Woodworth et al., 2018*). In contrast, transcripts encoding the low-affinity NTSR2 were present essentially in astrocytes (*Woodworth et al., 2018*), as we also recently demonstrated in the hippocampus (*Kyriatzis et al., 2021*). Studies with NTSR1$^{-/-}$ and NTSR2$^{-/-}$ mice and the absence of hypothermia effects of NTSR2-selective analogues suggested that in the different pathophysiological conditions evoked above, hypothermia is mediated by NTSR1 (*Pettibone et al., 2002*; *Remaury et al., 2002*; *Mechanic et al., 2009*; *Boules et al., 2010*). However, this remains controversial since receptor knockdown using antisense oligodeoxynucleotides in adult mice point to an involvement of NTSR2 in hypothermia (*Dubuc et al., 1999*). More recently, it appeared that activation of both NTSR1 and NTSR2 was required for a full hypothermic response (*Tabarean, 2020*).

## VH-N412 preserved learning and memory abilities, exploratory and normal locomotor activity in the KA model of epilepsy

Using two behavioral tasks, we report here that treatment of mice with VH-N412 after SE preserved learning and memory abilities as well as exploratory and general spontaneous motor activity. Hippocampus-dependent learning and memory performance was assessed using the OTM, in which mice were expected to make odor-reward associations in darkness, without visual cues (*Roman et al., 2002*). This test was selected in regard to the sensorial abilities of the FVB/N strain that are homozygous for the Pde6brd1 allele with an early onset retinal degeneration and blindness (*Brown and Wong, 2007*). Consequently, they are deeply impaired in spatial tasks such as the Morris water maze (*Brown and Wong, 2007*; *Owen et al., 1997*; *Pugh et al., 2004*; *Royle et al., 1999*; *Võikar et al., 2001*) or radial maze (*Mineur and Crusio, 2002*). In the OTM, it has already been demonstrated that albino mouse strains with reduced visual abilities such as BALB/C or CD1 mice were able to learn the task as well as or better than mouse strains with functional vision (*Restivo et al., 2006*; *Roman et al., 2002*). FVB/N mice reached a level of 80 ± 5% of correct responses from the fourth 12-trial training session. It has been shown that bilateral excitotoxic lesions of the hippocampus induced by injections of ibotenic acid, a glutamate agonist, led to learning and memory impairments of BALB/C mice in the OTM (*Nivet et al., 2011*). Using the FVB/N mice, similar learning and memory deficits were observed after injections of KA and consecutive seizures. In SE mice treated with VH-N412, we observed similar learning and memory performance compared to control mice. Similarly, behavioral activity was preserved in spontaneous locomotor activity using the open field test. SE and related hippocampal excitotoxic lesions are known to induce hyperactivity in rats (*dos Santos et al., 2000*; *Kubová et al., 2004*) and mice (*Chen et al., 2002*; *Müller et al., 2009*), which can be easily highlighted using the open field test. As expected, our experiments demonstrated a strong hyperactivity in FVB/N mice 4 weeks after KA injections and SE, that was not observed in animals treated with VH-N412.

## Cellular and molecular mechanisms potentially modulated by VH-N412 in the KA model of epilepsy

NT, its analogs, agonists, and antagonists have been shown to modulate both GABAergic and glutamatergic activity (*Ferraro et al., 2008*; *Li et al., 2008*). We investigated whether VH-N412 could modulate hippocampal neuronal hyperactivity induced by KA using MAE on acute hippocampal slices, but this was not the case. Our observations suggested that VH-N412 did not modulate neuronal hyperactivity in vivo, at least in the hippocampus. Modulators of the NT system may regulate neuroprotective pathways that are probably common to the different models of brain pathology evoked above. These models displayed neuronal excitotoxicity (reviewed in *Mattson, 2017*), neuroinflammation, as shown by pro-inflammatory glial markers such as astrocytic GFAP or microglial Iba1, endothelial and BBB damage (*Kyriatzis et al., 2021*; *Kyriatzis et al., 2024*). It has been shown in different models that PIH increased BDNF and vascular endothelial growth factor, reduced the pro-apoptotic caspase-3 activation, BAX, and MMP-9 expression, decreased expression of inflammatory factors

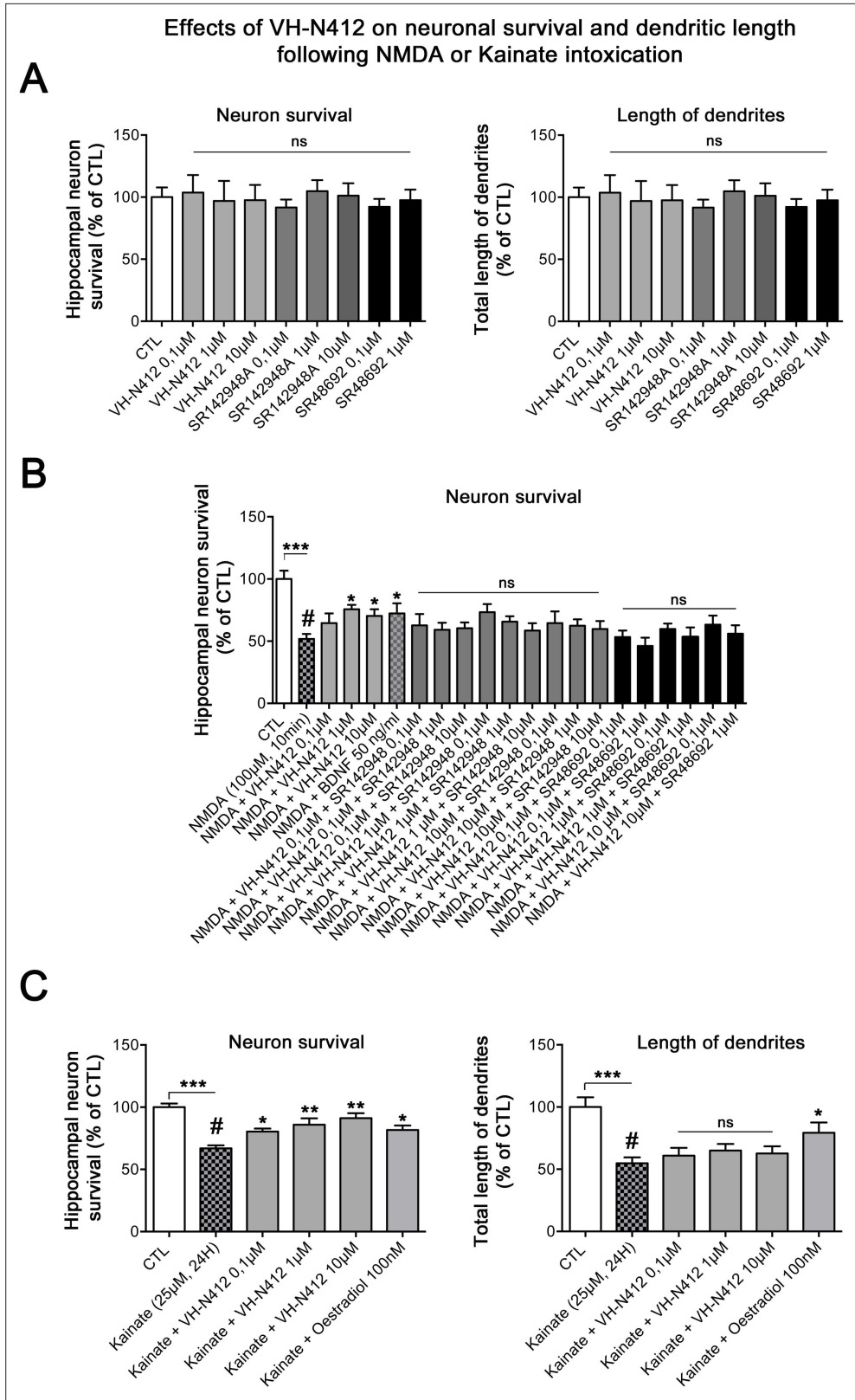

**Figure 13.** In vitro effects of VH-N412 on hippocampal neuronal survival and total dendrite length following NMDA or KA intoxication. (**A**) Histograms illustrate the effects of VH-N412 and two NTSR antagonists, SR142948A and SR48692, on hippocampal neuronal survival (left histogram) and on total dendrite length (right histogram). These compounds alone showed no toxic effects on hippocampal neuronal survival and on total dendrite length at

*Figure 13 continued on next page*

*Figure 13 continued*

all concentrations used. (**B**) Illustrates the effects of VH-N412 on survival of primary hippocampal neurons injured by NMDA. VH-N412 promoted neuronal survival at 1 and 10 μM with similar potency as that of BDNF 50 ng/mL. Antagonizing NTSR by SR142948A and SR48692 blocks the neuroprotective effect of VH-N412 in the neuronal cultures. (**C**) Illustrates the effects of VH-N412 on hippocampal neuronal survival (left panel) and on total dendrite length (right panel) following KA intoxication (25 μM). VH-N412 promoted neuronal survival at all concentrations (0.1, 1, and 10 μM) with similar potency as that of oestradiol at 100 nM. Following KA injury, VH-N412 did not display any significant effects on the total length of dendrites while oestradiol did.

including monocyte chemoattractant protein-1 and macrophage inflammatory protein-1α, two key chemokines in the regulation of microglia activation and infiltration. Moreover, upregulation of the M1 microglia markers interleukins-12 and -23, inducible nitric oxide synthase, tumor necrosis factor-α, interleukin-1β and -6 was also shown to be decreased or prevented. Meanwhile, TH increased the expression of M2-type reactive anti-inflammatory factors including interleukin-10, Fizz1, Ym1, and arginase-1 and the expression of the anti-apoptotic Bcl-2 (*Choi et al., 2012*; *Wei et al., 2013*; *Lee et al., 2014*; *Gu et al., 2015*; *Lee et al., 2016b*; *Jiang et al., 2017*; *Zhao et al., 2020*).

## Cold shock proteins RBM3 and CIRBP were involved in VH-N412-induced PIH in the KA model of epilepsy

Among the proteins known to be modulated by hypothermia are cold shock proteins such as RBM3 and CIRBP. They are known to escape translational repression, and are involved in diverse physiological and pathological processes, including hibernation, circadian rhythm, inflammation, neural plasticity, and they modulate neurodegeneration in relation with body temperature (*Williams et al., 2005*; *Smart et al., 2007*; *Chip et al., 2011*; *Tong et al., 2013*; reviewed in *Zhu et al., 2016*; *Ávila-Gómez et al., 2020*).

Following VH-N412 administration that induced hypothermia in naïve mice, we observed significant increase of mRNA encoding RBM3 and CIRBP in hippocampus relative to control at 16 and 8hr time points, respectively, which correlated with RBM3 and CIRBP protein increase. In agreement with previous studies (*Peretti et al., 2015*), the absence of modulation of RBM3 and CIRBP mRNA we observed in neurons cultured at 37°C and treated with different concentrations of VH-N412 suggests that it is indeed hypothermia that induced expression of these cold shock proteins in our epileptic model. Induction of these cold shock proteins in epileptic mice treated with VH-N412 is probably associated with the observed neuroprotection as shown in other experimental paradigms (*Peretti et al., 2015*).

## VH-N412 had intrinsic neuroprotective effects in cultured hippocampal neurons, independent of its potential to induce hypothermia

The hippocampus is known as a highly vulnerable structure of the CNS, with rapid loss of principle cells including pyramidal cells and interneurons (reviewed in *Houser, 2014*). Remarkably, these neurons were protected by VH-N412 administration in our epilepsy model. Two processes may be involved in such neuroprotection: (i) either hypothermia with a global reduction of cell metabolism, oxidative stress, and excitotoxic processes, with concomitant reduction of deleterious neuroinflammation; (ii) intrinsic neuroprotective effects of VH-N412 that could act in synergy with hypothermia. These two processes are not exclusive and it is possible that the conjugate causes hypothermia and has favorable effects on the sequelae of SE. Further experiments are warranted where one could prevent pharmacological hypothermia by warming up the animals undergoing SE and administered with VH-N412, to evaluate the intrinsic neuroprotective effects of the conjugate in vivo.

Our immunohistochemistry experiments on mouse brain sections showed that hippocampal neurons expressed the NTSR1 receptor, that is located not only in the neuronal cell bodies, but also in proximal dendrites and in spines. This was confirmed in cultured hippocampal neurons expressing either the endogenous or exogenous NTSR1. These observations lead us to question whether VH-N412 displayed neuroprotective effects on pyramidal neurons in culture. These were challenged with two excitotoxic agents, NMDA and KA, that induce excitotoxic neuronal death via their respective receptors. We showed that VH-N412 did induce neuroprotection as efficiently as molecules well known for neuroprotection, BDNF, and oestradiol (*Mattson et al., 1995*; *Kajta and*

*Lasoń, 2000*). This is in agreement with data showing that the neurotoxic effects of methamphetamine on dopaminergic terminals and apoptosis of striatal neurons were attenuated by the NTSR1 agonist PD149163, independently of hypothermia (*Liu et al., 2017*). In another study, exogenous treatment with the PD149163 NTSR1 agonist exerted neuroprotection in HFD-induced prediabetic rats and Alzheimer models. It was hypothesized that the treatment with PD149163 attenuates hippocampal pathologies and synaptic dysfunction, eventually restoring cognition through hippocampal NTSR1 signaling. In particular, it was shown that the NTSR1 agonist PD149163 increased dendritic spines and reduced hippocampal microglial cells (*Saiyasit et al., 2021*). Finally, as mentioned above, NTSR agonists induced neuroprotection in a number of experimental conditions including global ischemia, stroke, TBI, etc. At this point it is difficult to determine the direct contribution of hypothermia compared to intrinsic neuroprotective effects of these agonists. In contrast, other studies point to the deleterious effects of NTSR agonists on glutamatergic excitotoxicity both in vitro and in vivo. NT enhanced glutamate-induced excitotoxicity through NTSR1 activation in both mesencephalic and cortical neurons (*Antonelli et al., 2004*; *Antonelli et al., 2002*). NT increased glutamate release in some brain regions (*Sanz et al., 1993*; *Ferraro et al., 1995*) and it has been shown that in addition to enhancing glutamate release, NT can modify the function of glutamate receptors in vitro and in vivo (*Antonelli et al., 2004*; *Ferraro et al., 2011*; *Ferraro et al., 2008*).

## Conclusion

In all, our results suggested that if indeed VH-N412 exerted neuroprotective effects via hypothermia in our epileptic model, there was some in vitro evidence that this molecule exhibited neuroprotective effects independently from hypothermia. This hypothesis will need to be challenged further in vitro and in other models of brain diseases in vivo. In the present article, we essentially focused on epilepsy and acute neurotoxicity models and showed the beneficial effects of a vectorized active peptide such as NT following systemic administration. The CNS is endowed with many devastating pathologies including neurodegenerative diseases. The latter could benefit from new drugs, some of them in development, including biologics such as oligonucleotides, RNAi, proteins, or therapeutic antibodies, at the condition they reach the diseased CNS. Thus, our results emphasize the potential of drug delivery approaches based on targeting peptide or antibody fragment moieties to bring conjugated drugs across the BBB into the diseased CNS, following systemic administration of conjugates.

## Materials and methods

### Animals

All animals were housed 6 per cage in a temperature and humidity-controlled room (22 ± 2°C, 12 hr light-dark cycles), had free access to food and water. Procedures involving animals were carried out according to National and European regulations (EU Directive N°2010/63) and to authorizations delivered to our animal facility (N° C13 055 08) and to the project (N° 00757.02) by the French Ministry of Research and Local Ethics Committee. All efforts were made to minimize animal suffering and reduce the number of animals used. Hypothermia was evaluated in 5-week-old male Swiss CD-1 mice (Janvier Laboratories, Le Genest-Saint-Isle, France) using a digital thermometer rectal probe (Physitemp Intruments Inc, Clifton, NJ, USA). Peptide stability was assessed in fresh mouse blood collected from Swiss CD-1 mice. For the KA model and for behavioral tests (see below), young adult male FVB/N mice (Janvier Laboratories) were used. The FVB/N mice are reliable and well-described mouse models of epilepsy, where seizures are associated with cell death and inflammation (*Schauwecker, 2003*; *Wu et al., 2021*).

### Peptide synthesis and characterization

All commercially available reagents and solvents were used as received without further purification. Resins (Fmoc-Gly-Wang Resin [100–200 mesh, 1% DVB, loading 0.7 mmol/g] and Fmoc-Leu-Wang Resin [100–200 mesh, 1% DVB, loading 0.7 mmol/g]) were purchased from Iris Biotech (Marktredwitz, Germany). Common chemical reagents were purchased from Bachem AG (Bubendorf, Switzerland), Sigma-Aldrich (St. Louis, MO, USA), Analytic Lab (St-Mathieu de Tréviers, France), Polypeptide Laboratories (Strasbourg, France), or Thermo Fisher Scientific (Waltham, MA, USA). Mass spectrometry (MS)

data were registered using a MALDI-TOF mass spectrometer (MALDI-TOF-TOF Ultraflex II Bruker, Mundelein, IL, USA) in positive mode. Internal calibration was applied by using 4-HCCA matrix $m/z$.

Reaction progress and purity monitoring were carried out using analytical RP-HPLC separation with a Dionex UltiMate 3000 system (Thermo Fisher Scientific) equipped with a C18 Kinetex (5 µm, 150 mm × 4.6 mm). Detection was performed at 214 nm. Elution system was composed of $H_2O$/0.1% TFA (solution A) and MeCN/0.1% TFA (solution B). Flow rate was 2 mL/min with a gradient of 0–100% B in 4 min for reaction monitoring and 0–100% B in 10 min for purity assessment. Retention times ($R_t$) from analytical RP-HPLC are reported in min. Crude products were purified by RP-HPLC with a Dionex Ulti-Mate 3000 system equipped with a C18 Luna (5 µm, 100 mm × 21.2 mm). Detection was performed at 214 nm. Elution buffer was composed of $H_2O$/0.1% TFA (solution A) and MeCN/0.1% TFA (solution B). Flow rate was 20 mL/min.

## Synthesis of VH-N21 and VH-N41

### Synthesis of Ac-[cMPRLRGC]c-G-OH

Peptide Ac-cMPRLRGC-G-OH (VH445) was synthesized by solid phase peptide synthesis (SPPS) method on a CEM Liberty microwave synthesizer (Matthews, NC, USA), using standard Fmoc/tBu strategy and a Fmoc-Gly-Wang Resin (100–200 mesh, 1% DVB, loading 0.7 mmol/g) purchased from Iris Biotech on a 0.25 mmol scale. Fmoc deprotection was performed via microwave irradiation with piperidin 20% in DMF. Amino acids were coupled via microwave activation using the mixture aa/DIEA/HATU: 4/4/8 equivalent with respect to the resin. Coupling time was adjusted to 10 min. Double couplings were necessary for methionine and cysteine incorporation after the proline residue. N-terminal acetylation was performed manually using 50% acetic anhydride in DCM, 2×5 min at room temperature (RT) with gentle stirring. Resin-bound peptide was then cleaved using a solution comprised of TFA/TIS/$H_2O$/EDT: 94/2/2/2 at RT for 2 hr. A minimum of 15 mL of cleavage solution was used per gram of resin. Crude peptide was then precipitated using ice-cold ether, centrifuged at 3000 rpm for 8 min and lyophilized in $H_2O$/0.1% TFA to obtain a white powder. Crude Ac-cMPRLRGC-G-OH peptide was then dissolved in AcOH 0.5% for a final concentration of 0.5 mg/mL. Ammonium carbonate (2N) was added to the peptide solution to reach an approximate basic pH of 8–9. $K_3[Fe(CN)_6]$ (0.01N) was then added to the reaction mixture until a bright and persistent yellow color was observed. Monitoring of the reaction was performed by analytical RP-HPLC. After 30 min, the reaction mixture was purified by preparative RP-HPLC ($R_t$=6 min, C18 Luna, 5 µm, 100 mm × 21.2 mm, flow rate = 20 mL/min, phase A was 0.1% TFA in water, phase B was 0.1% TFA in acetonitrile, gradient was linear from 14% to 24% B over 30 min) to generate Ac-[cMPRLRGC]c-G-OH as a pure white powder (35.3 mg, 14% yield, purity >95%). $R_t$ = 1.73 min, C18 Kinetex, 5 µm, 150 mm × 4.6 mm, flow rate = 2 mL/min, phase A was 0.1% TFA in water, phase B was 0.1% TFA in acetonitrile, gradient was linear from 0% to 100% B over 4 min. MALDI-TOF ($m/z$) for $C_{40}H_{69}N_{15}O_{11}S_3$, $[M+H]^+$ calc. 1032.40 Da, found 1032.42 Da.

### Synthesis of Ac-[cMPRLRGC]c-G-CH2-CH2-SH

Trt-cysteamine (2 eq.), PyBop (1.1 eq), and DIEA (4 eq.) were added to a 0.05 M solution of Ac-[cMPRLRGC]c-G-OH (28 mg) in anhydrous DMF. The reaction completion was monitored by RP-HPLC. The analysis showed the complete disappearance of the starting peptide after 5 min. DMF was then evaporated under vacuum to yield protected Ac-[cMPRLRGC]c-G-CH2-CH2-STrt as a pale-yellow oil. Trityl protection of the sulfur was removed by dissolving the crude oil in a solution of DCM/TIS/TFA: 3/1/1 (0.5 mL/mg of peptide). Reaction mixture was allowed to stir at RT. Monitoring of the reaction was performed by analytical RP-HPLC. DCM and TFA were then evaporated using inert gas bubbling and the crude peptide was subsequently precipitated with excess volume of ice-cold ether. After centrifugation at 3000 rpm for 8 min crude peptides were lyophilized in $H_2O$/0.1% TFA and purified by preparative RP-HPLC ($R_t$ = 16 min, C18 Luna, 5 µm, 100 mm × 4.6 mm, flow rate = 20 mL/min, phase A was 0.1% TFA in water, phase B was 0.1% TFA in acetonitrile, gradient was linear from 12% to 22% B over 30 min) to generate Ac-[cMPRLRGC]c-G-CH2-CH2-SH as a pure white powder (16 mg, 54% yield, purity>95%). $R_t$ = 1.81 min (C18 Kinetex, 5 µm, 150 mm × 4.6 mm, flow rate = 2 mL/min, phase A was 0.1% TFA in water, phase B was 0.1% TFA in acetonitrile, gradient was linear from 0% to 100% B over 4 min). MALDI-TOF ($m/z$) for $C_{42}H_{74}N_{16}O_{10}S_4$, $[M+H]^+$ calc. 1091.40 Da, found 1091.44 Da.

## Synthesis of [Lys(MHA)6]NT

A solution of sulfo-EMCS (12.2 mg, 29.8 µmol, 1 eq.) in 2 mL of PBS 4× (pH = 7.4) was added to a solution of NT (pELYENKPRRPYIL-OH, 50 mg, 29.8 µmol, 1 eq.) in 7 mL PBS 4× (pH = 7.6). Reaction mixture was allowed to stir at RT. Monitoring of the reaction was performed by analytical RP-HPLC. After overnight stirring the crude mixture was purified by preparative RP-HPLC ($R_t$ = 15 min, C18 Luna, 5 µm, 100 mm × 21.2 mm, flow rate = 20 mL/min, phase A was 0.1% TFA in water, phase B was 0.1% TFA in acetonitrile, gradient was linear from 21% to 31% B over 30 min) to generate [Lys(MHA)6]NT as a pure white powder (m=25 mg, 45% yield, purity >95%). $R_t$ = 2.15 min (C18 Kinetex, 5 µm, 150 mm × 4.6 mm, flow rate = 2 mL/min, phase A was 0.1% TFA in water, phase B was 0.1% TFA in acetonitrile, gradient was linear from 0% to 100% B over 4 min). MALDI-TOF (*m/z*) for $C_{88}H_{133}N_{22}O_{23}$, $[M+H]^+$ calc. 1865.98 Da, found 1865.97 Da.

## Conjugation of [Lys(MHA)6]NT to Ac-[cMPRLRGC]c-G-CH2-CH2-SH

[Lys(MHA)6]NT (1 eq.) was added to a solution of Ac-[cMPRLRGC]$_c$-G-CH$_2$-CH$_2$-SH in PBS 1× (1 eq., 0.003 M, pH 7.4). The reaction mixture was allowed to stir at RT. Monitoring of the reaction was performed by analytical RP-HPLC. After reaction completion the mixture was purified by preparative RP-HPLC ($R_t$ = 23 min, C18 Luna, 5 µm, 100 mm × 21.2 mm, flow rate = 20 mL/min, phase A was 0.1% TFA in water, phase B was 0.1% TFA in acetonitrile, gradient was linear from 20% to 30% B over 40 min) to generate VH-N21 as a pure white powder (34% yield, purity >95%). $R_t$ = 2.11 min (C18 Kinetex, 5 µm, 150 mm × 4.6 mm, flow rate = 2 mL/min, phase A was 0.1% TFA in water, phase B was 0.1% TFA in acetonitrile, gradient was linear from 0% to 100% B over 4 min). MALDI-TOF (*m/z*) for $C_{130}H_{206}N_{38}O_{33}S_4$, $[M+H]^+$ calc. 2956.30, found 2956.67 Da. The molecule resulting from the conjugation of [Lys(MHA)6]NT to Ac-[cMPRLRGC]$_c$-G-CH$_2$-CH$_2$-SH was named VH-N21. Another conjugate was also synthesized based on the conjugation of [Lys(MHA)6]NT to the previously described LDLR-binding peptide VH4129 [cM"Pip"RLR"Sar"C]$_c$ (*Jacquot et al., 2016*), with the exception that the N-ter was propionylated rather than acetylated. This conjugate was named VH-N41.

## Synthesis of VH-N412

VH-N412 was synthesized by SPPS method using standard Fmoc/tBu strategy and a Fmoc-Leu-Wang Resin (100–200 mesh, 1% DVB, loading 0.7 mmol/g) purchased from Iris Biotech on a 0.25 mmol scale. Synthesis was performed on a CEM Liberty microwave synthesizer except for the Fmoc-21-amino-4,7,10,13,16,19-hexaoxaheneicosanoic acid that was introduced manually overnight using COMU/DIEA (2/2/8 eq. with respect to the resin) in anhydrous DMF (0.2 M) as coupling cocktail. For automated synthesis Fmoc deprotection was performed via microwave irradiation with piperidin 20% in DMF. Amino acids were coupled via microwave activation using the mixture aa/DIEA/HATU: 4/4/8 equivalent with respect to the resin. Coupling time was adjusted to 10 min. Double couplings were necessary for methionine and cysteine incorporation after the proline residue in the VH moiety. N-terminal propionylation was performed manually using 50% propionic anhydride in DCM (2×5 min at RT with gentle stirring). Resin-bound peptide was then cleaved using a solution comprised of TFA/TIS/H2O/EDT: 94/2/2/2 for 2 hr at RT. A minimum of 15 mL of cleavage solution was used per gram of resin. Crude peptide was then precipitated using ice-cold ether, centrifuged at 3000 rpm for 8 min, and lyophilized in H$_2$O/0.1% TFA to obtain a white powder. Crude Pr-cMPRLRGC-PEG6-RRPYIL-OH peptide was then dissolved in AcOH 0.5% to reach 0.5 mg/mL final concentration. Ammonium carbonate (2N) was added to the peptide solution to reach an approximate basic pH of 8–9. K$_3$[Fe(CN)$_6$] (0.01N) was then added to the reaction mixture until a bright and persistent yellow color was observed. Monitoring of the reaction was performed by analytical RP-HPLC. After 30 min, the reaction mixture was purified by preparative RP-HPLC ($R_t$ = 18 min, C18 Luna, 5 µm, 100 mm × 21.2 mm, flow rate = 20 mL/min, phase A was 0.1% TFA in water, phase B was 0.1% TFA in acetonitrile, gradient was linear from 19% to 29% B over 30 min) to generate Pr-[cMPRLRGC]$_c$-PEG6-RRPYIL-OH as a pure white to yellow powder (73.5 mg, 14% yield, purity >95%). $R_t$ = 4.34 min (C18 Kinetex, 5 µm, 150 mm × 4.6 mm, flow rate = 2 mL/min, phase A was 0.1% TFA in water, phase B was 0.1% TFA in acetonitrile, gradient was linear from 0% to 100% B over 10 min). MALDI-TOF (*m/z*) for $C_{94}H_{163}N_{27}O_{24}S_3$, $[M+H]^+$ calc. 2151.16, found 2151.11 Da.

## LDLR binding using SPR analysis

Recombinant human LDLR (His-tagged) was purchased from Sino Biological (Beijing, China). Interaction of ligands with hLDLR was tested at 25°C using a Biacore T200 apparatus (GE Healthcare, Buc, France) and HBS (50 mM HEPES-NaOH pH 7.4, 150 mM NaCl, 0.005% Tween-20, 50 μM EDTA) as running buffer. hLDLR was immobilized on a NiHC sensor chip (Xantec, Dusseldorf, Germany) at a density of around 25 pmol/mm². Binding to hLDLR-coated flow cells was corrected for nonspecific binding to uncoated flow cell. The single-cycle kinetic method was used to measure the affinity of the VH4129 peptide and vectorized NT (VH-N412) for hLDLR. VH4129 and VH-N412 were serially diluted twofold in running buffer yielding concentrations ranging from 50 to 800 nM and samples were injected sequentially every 2 min at 30 μL/min using increasing concentrations. Blank run injections of HBS were performed in the same conditions before injection of the ligands. Double-subtracted sensorgrams were globally fitted with the 1:1 binding model from Biacore T200 Evaluation version 2.0. Data are representative of at least three independent experiments.

## Binding to human and rat NTSR1 using a competition assay on NTSR1-expressing membrane extracts

The binding affinity of the NT conjugates was assessed using a competition binding assay using rat and human NTSR1 membrane homogenates. The cell membranes enriched for the human and rat NTSR1 membranes were prepared as follows: 1 day before transfection, HEK 293 cells (ECACC, Salisbury, UK) were seeded and cells were transfected using Lipofectamine 2000 (Thermo Fisher Scientific) with the rat and human NTSR1 plasmid constructs (Origene, Rockville, MD, USA). Cell expression of the rat NTSR1 receptor was evaluated using western blot and immunocytochemistry as described in detail below. At 100% confluence, the medium was removed, and the cells were harvested using PBS EDTA 5 mM buffer. They were washed with ice-cold PBS and centrifuged at 1700×$g$ for 10 min at 4°C. The pellet was suspended in ice-cold buffer (1 mM EDTA, 25 mM sodium phosphate, 5 mM MgCl$_2$, pH 7.4) and homogenized using a Potter-Elvehjem homogenizer (Fisher Scientific, Elancourt, France). The homogenate was centrifuged at 1700× $g$ for 15 min (4°C). The pellet was washed, resuspended, and centrifuged at 1700×$g$ for 15 min (4°C). The combined supernatants were centrifuged at 100,000×$g$ for 40 min (4°C) on a TL110 rotor (Beckman Coulter, Paris, France) and the pellet was suspended in the same buffer. Protein concentrations were determined using the DC protein assay kit (Bio-Rad, Hercules, CA, USA). The membrane preparations were aliquoted and stored at –80°C. Assays on hNTSR-1 and rNTSR-1 were performed using membrane homogenates at a final concentration of 0.5 μg/well or 0.75 μg/well, respectively, and the radioligand [³H]-NT (specific activity 99.8 Ci/mmol; Perkin-Elmer) at a concentration of 3 nM. Specific binding of the radioligand was determined with $K_D$ values of 0.81 nM and a $B_{max}$ of 34 pmol per mg membrane protein for binding to hNTSR-1 and $K_D$ values of 0.93 nM and a $B_{max}$ of 2.5 pmol per mg membrane protein for binding to hNTSR-1. Nonspecific binding was determined in the presence of 5 μM NT. Each assay was performed in a 96-well plate in a total reaction volume of 100 μL in binding buffer (50 mM Tris-HCl pH 7.4, 0.1% bovine serum albumin [BSA, Sigma]). The assay was subsequently incubated for 24 hr at RT. The content of each well was rapidly filtered through Unifliter-96, (GF/C filters [Perkin-Elmer] presoaked with 25 μL of 0.5% polyethylenimine) using a MicroBeta FilterMate-96 Harvester (Perkin-Elmer) and each well was rinsed five times with washing buffer (10 mM Tris-HCl, pH 7.2). Radioactivity (cpm) of each dried filter was measured by adding 25 μL of MicroScint-O (Perkin-Elmer) and quantified using TopCount NXT Microplate Scintillation and Luminescence Counter (Perkin-Elmer). Specific binding was typically 90% or greater of the total binding. Dose-response curves were plotted using KaleidaGraph to determine IC50 values. Assays were performed in duplicate. Ki values were determined from mean IC50s using the Cheng-Prusoff conversion.

## In vitro stability of conjugates in freshly collected blood

In vitro blood stability (half-life or $t_{1/2}$) of the VH-N21 and VH-N412 conjugates was assessed and compared to that of NT alone as previously performed with free VH445 analogues (*Malcor et al., 2012*; *Jacquot et al., 2016*). Briefly, each peptide was incubated at the nominal concentration of 2 μM up to 3 hr at 37°C in freshly collected lithium heparin blood from Swiss CD-1 mouse or human. The analyte was quantified in the plasma fraction at several time points by liquid chromatography-tandem mass spectrometry, using a Shimadzu LC equipment coupled to an API 4000 triple quadrupole mass

spectrometer (Applied Biosystems, Foster City, CA, USA). In vitro half-life ($t_{1/2}$) was estimated from the logarithmic regression of each kinetic profile (first-order reaction kinetics: $C(t)=C_0 \cdot e^{-kt}$) and given by $t_{1/2}=\ln 2/k$.

## Radiolabeling of vectorized and non-vectorized NT and in situ brain perfusion in mice

Radiolabeled activated NT compounds were obtained either by coupling [3H]Tyr3-NT with the LDLR-binding peptide or by coupling of a tritiated LDLR-binding peptide to NT molecules. NT was radiolabeled by palladium-catalyzed dehalogenation of 3,5-diiodoTyr3-NT using tritium gas and Pd/C while the LDLR-binding peptide was radiolabeled by coupling of a tritiated propionyl succinimide ester to its N-terminus. The specific radioactivity was typically in the range of 50 Ci/mmol. The total quantity of radioactivity prepared for each synthesis was generally between 100 and 1000 μCi.

BBB transport of the NT and NT-peptide conjugates was assessed in adult male mice as described in detail in *Dagenais et al., 2000*. Briefly, the in situ brain perfusion of the radiolabeled compounds was performed during 120 s with a perfusate flow rate of 2.5 mL/min. The initial transport was expressed as the brain clearance $K_{in}$ and corrected for the vascular volume with $^{14}C$-sucrose co-perfusion. $^{14}C$-sucrose, a compound that does not cross the intact BBB, allowed assessment of its physical integrity. The BBB transport of the radiolabeled peptides was expressed as the $K_{in}$ $(\mu L/s/g)=V_{brain}/T$ (where $V_{brain}$ is the apparent volume of distribution of the tritium-labeled compound corrected from vascular contamination, where T is the perfusion time).

## KA model of temporal lobe epilepsy

Forty-eight young adult male FVB/N mice (25–30 g; 9–10 weeks of age) were injected s.c. with a single dose of KA (45 mg/kg; Abcam, Cambridge, UK) to generate mice with spontaneous recurrent seizures as a hallmark of SE, as previously described (*Schauwecker and Steward, 1997*; *Gröticke et al., 2008*; *Wu et al., 2021*). KA-injected mice were individually housed and received a 0.5 mL i.p. dose of glucose G5 and had free access to agarose-doliprane gel (2.4%; Sanofi, Gentilly, France) to avoid pain. Mice were observed during 9 hr for onset and extent of seizure activity. Seizure activity was scored by visual inspection according to a modified *Racine, 1972*, scale: stage 0: exploration, 'normal behavior'; stage 1: immobility, staring; stage 2: head nodding and/or extended tail; stage 3: forelimb clonus and/or circling behavior; stage 4: rearing with forelimb clonus and falling; stage 5: continuous rearing and falling; stage 6: severe tonic-clonic seizures. Only animals having displayed at least stage 5 were included in the study and five animal groups were generated: (i) animals injected 30 min after the onset of the SE with one i.v. (tail vein) bolus injection of the VH-N412 compound at the dose of 4 mg/kg eq. NT (group 'SE30+VH-N412', n=13); (ii) animals injected 30 min after the onset of the SE with an i.v. bolus injection of NT at the dose of 4 mg/kg eq. NT (group 'SE30+NT(8–13)', n=14); (iii) animals injected 30 min after the onset of the SE with DZP i.p. (Valium, Roche, Basel, Switzerland) at the dose of 15 mg/kg (group 'SE30+DZP', n=5); (iv) animals injected 30 min after the onset of the SE with saline 0.9% (vehicle control) (group 'SE', n=11); (v) a negative control group of age-matched mice was administered saline 0.9% alone (group 'SHAM', n=5). All animals were weighed and monitored daily. Body temperature was assessed using a digital thermometer rectal probe before KA injection and every 30 min during 3 hr thereafter. Seizure intensity score was assessed following KA injection and every 30 min during 3 hr. As SE did not occur at the same time for each animal, recordings were performed from the onset of SE for comparative analysis of changes in body temperature and for seizure intensity score.

To study mice in the chronic stage of epilepsy with spontaneous seizures, they were observed daily (at least 3 hr per day) for general behavior and occurrence of SRS. These are highly reproducible in the mouse KA model, allowing for visual monitoring and scoring of epileptic activity. After 3 weeks, most animals exhibited SRS, with two to three seizures per day, similar to previous observations (*Wu et al., 2021*). The detection of at least one spontaneous seizure per day was used as criterion indicating the animals had reached chronic phase that was ultimately confirmed by mossy fiber sprouting.

## Tissue preparation

Mice were deeply anesthetized with pentobarbital (700 mg/kg; Sigma) and transcardially perfused with 4% paraformaldehyde. The brains were extracted and post-fixed for 1 hr at RT and rinsed in

0.12 M phosphate buffer (0.12 M PB; pH 7.4). Forty µm coronal sections were cut on a vibratome, immersed in a cryoprotective solution, and stored at –20°C until histological assessment. Every tenth section was stained with cresyl violet to determine the general histological characteristics of the tissue within the rostro-caudal extent of the brain. From each mouse, selected sections from the dorsal hippocampus were then processed for tissue evaluation by using NeuN immunohistochemistry or FJC staining. Adjacent sections were treated for assessment of inflammation and mossy fiber sprouting. Histopathological analyses were performed at two post-SE intervals: at 1 week to estimate hippocampal sclerosis and at 2 months to assess mossy fiber sprouting. For each experiment, sections from the different animal groups were processed in parallel.

## FJC staining and scoring of neurodegeneration

To estimate cell degeneration, we performed FJC histo-staining (Sigma) on sections of dorsal hippocampus from SHAM, SE, SE+VH-N412, SE+NT(8–13), and SE+DZP animals according to the standard protocol of *Schmued et al., 2005*, and *Bian et al., 2007*. This histostaining does not depend on the mode of cell death (*Ikenari et al., 2021*). The specimens were analyzed using a confocal microscope and images were acquired using Zen software (Zeiss, LSM 700, Jena Germany) and processed using Adobe Photoshop and ImageJ softwares. The mean fluorescence intensity of FJC (mean gray value) was measured in Z-stack ×10 z1 acquired images of pyramidal cell layers CA1, CA3, H, GCL. A minimum of four slices were analyzed on both sides of the dorsal hippocampus for each mouse of each group. Values were obtained following analysis of n=3–4 mice per group. We subtracted fluorescence value in areas devoid of stained cells in the same sections; an average value of 4 for each area was obtained from every image. Data are presented as the mean percentage of fluorescence of FJC normalized to that of SHAM (mean ± SEM).

To confirm the neuronal identity of degenerating cells and to evaluate the degree of neurodegeneration, we scored NeuN and FJC staining on hippocampal sections. The score ranged from 0 (absence of neuronal death or FJC positive neurons) to 7 (maximal neurodegeneration). A score of 1 was given for positive neurons either in the H, in the GCL, in the CA3, CA1, or CA2 pyramidal cell layers. Whenever an intense neurodegeneration was observed in the CA3 or CA1 pyramidal cell layer (many positive cells with confluent fluorescence), the score related to this area was increased by 1. The scores obtained on NeuN and FJC sections were adjusted. The numbers resulting from these values were averaged per animal and used for statistical analysis.

## Immunohistochemical labeling

For immunofluorescence labeling, dorsal hippocampal sections from SHAM, SE, SE+VH-N412, SE+NT(8–13), and SE+DZP animals were preincubated for 1 hr at RT in 0.12 M PB containing 0.3% Triton X-100 and 3% BSA. Sections were then incubated overnight at 4°C in a solution containing the primary antibody: mouse anti-NeuN (1/1000; MAB377), or mouse anti-GFAP (1/1000, MAB360), both from Millipore, Darmstadt, Germany, or rabbit anti-ionized calcium binding adaptor molecule 1 (Iba1; 1/1000, Wako Pure Chemical Industries, Osaka, Japan), or rabbit anti-vesicular zinc transporter 3 (ZnT-3; 1/500; Synaptic System, Goettingen, Germany) diluted in PB 0.12 M containing 0.3% Triton X-100 and 3% BSA. After several rinses, sections were incubated for 2 hr at RT in Alexa Fluor 488-conjugated anti-mouse or -rabbit IgG (1/200; Thermo Fisher Scientific) diluted in PB containing 3% BSA. Sections were then rinsed and coverslipped with Prolong Gold Anti-fading reagent (Life Technologies). In all immunohistochemical experiments, no labeling was detected when primary antibodies were omitted.

## Estimation of neuroinflammation

Astroglial (GFAP) and microglial (Iba1) reactivities were quantified by measuring labeling intensities on whole hippocampal sections from SHAM, SE, SE+VH-N412, SE+NT(8–13), and SE+DZP animals. The densitometric data were automatically calculated by ImageJ software. For each animal group the mean and corresponding SEM intensity of labeling obtained from the total number of microscopic fields were calculated.

## Estimation of mossy fiber sprouting

Quantitative analysis was conducted to evaluate the number of mossy fiber terminals present in the DG IML from SHAM, SE, SE+VH-N412, and SE+NT(8–13) animals. These mossy fiber terminals

revealed by ZnT-3 project aberrantly into this layer after KA-induced SE (*Victor Nadler et al., 1980*; *Sloviter et al., 2006*; *Epsztein et al., 2006*). For this purpose, five Z-stacks of three confocal slices were acquired regularly along the IML for each animal, with a ×40 objective and a numerical zoom 1.2. A region of interest (ROI) was outlined for each slice, and the terminals contained in the ROI were counted and averaged with the two other slices. The Cell Count plugin of ImageJ was used and counts were averaged to one final score per animal and mean numbers were used for statistical analysis.

## Behavioral analysis
### Olfactory tubing maze
The hippocampus-dependent learning and memory performance of control mice injected with saline solution (SHAM), SE, and SE+VH-N412 mice were assessed 3 weeks after SE (12-week-old male mice). Behavioral assessment, based on the OTM, was carried out as we previously described (*Roman et al., 2002*; *Girard et al., 2014*) and we have shown that the FVB/N mouse strain used in the present study performed very well in this task (*Girard et al., 2016*). Briefly, mice were trained to learn odor-reward associations. The mice placed on water restriction were submitted to the presentation of odors in testing chambers (TC). Two synthetic odors (strawberry and jasmine), coming from two distinct arms and randomly assigned to these arms, were simultaneously presented to the mice. Each of these two odors was arbitrarily associated either with a drop of water (the reward) or with a non-aversive but unpleasant sound, which were delivered depending on the arm chosen by the mice. The OTM was composed of four identical TC joined to each other forming a circular structure in which mice could move freely only in clockwise direction. The procedure was fully automated. The movement of the mice was detected by photoelectric cells. Odors, water, sound delivery, and the automatic doors separating the different TC were controlled by a microcomputer running with LabVIEW software (National Instruments, Austin, TX, USA) that also automatically recorded the behavioral data. The learning procedure included 3 habituation days followed by 5 days of training, each daily session being composed of 12 trials/odor presentations corresponding to 3 clockwise laps/blocks of 4 trials. Two parameters were examined: the ITI and the percentage of correct responses. The ITI was calculated as the time between the response to an odor presentation in one testing chamber and the response in the next chamber. The evolution of the ITI reflects the fact that, after a response to an odor, the animal must learn to backtrack to the testing chamber and to run to the next chamber waiting for the opening of the entrance door for the next trial. This constitutes a procedural aspect of the task. The percentage of correct responses were the ratio of the number of correct responses to the total number of odor presentations per session. This percentage was used to evaluate the effectiveness of the association between an odor and its reinforcement, a process that pertains to the hippocampus-dependent declarative memory subcategory.

### Open field test
As we described previously (*Girard et al., 2016*; *Girard et al., 2014*), the open field test was used to assess both exploratory behavior and spontaneous locomotor activity 4 weeks after SE and VH-N412 treatment. The mice were tested using an open field square made of white plastic with 50 cm×50 cm surface area and 45-cm-high walls. Testing was carried out in a dimly illuminated room (40 lux). Monitoring was done by an automated tracking system using a video camera mounted above the apparatus (Viewpoint VideoTrack version 3.0, Lyon, France). The field was divided virtually into two regions of interest: a central area (20 cm×20 cm) and a peripheral area. The animals were placed in the center of the field and two behavioral parameters were registered during two consecutive 5 min sessions: (i) the percentage of time spent in the central part vs total time and (ii) the total traveled distance. The square was cleaned prior to the test and between each animal with 70% ethanol.

## RT-qPCR and evaluation of cold shock protein mRNA levels
Total RNA was prepared from hippocampi of SHAM and VH-N412-treated mice for 4, 8, and 16 hr and from rat mature hippocampal neuronal cultures treated with VH-N412 at 0.1, 1, and 10 μM using the Nucleospin RNA plus kit (Macherey Nagel, Allentown, PA, USA). For the RT-qPCR experiments, all reagents, kits, equipment, and software were from Applied Biosystems. cDNA was synthesized from 500 ng of total RNA using the High-Capacity RNA-to-cDNA Kit. For real-time qPCR, 12.5 ng of cDNA were used. The samples were run in duplicate on 96-well plates and then analyzed with

7500 v2.0 software according to the manufacturer's recommendations. All reactions were performed using TaqMan Fast Universal PCR Mix and TaqMan Assay probes (*Supplementary file 4*). The conditions of the thermal cycle were as follows: initial denaturation at 95°C for 40 cycles, denaturation at 95°C, and hybridization and extension at 60°C. Relative expression levels were determined according to the $\Delta\Delta Ct$ (Ct: cycle threshold) method where the expression level of the mRNA of interest is given by $2^{-\Delta\Delta CT}$, where $\Delta\Delta CT = \Delta Ct$ target mRNA - $\Delta Ct$ reference mRNA (GAPDH, housekeeping gene) in the same sample.

## Immunoblot analysis of cold shock protein expression

Mouse brains were rapidly extracted, and the hippocampi from SHAM and VH-N412-treated animals were isolated and lysed in RIPA buffer (Sigma). After sonication, protein concentrations were determined using the Bio-Rad DC protein assay kit (Bio-Rad, Hercules, CA, USA). Protein extracts (100 µg) were loaded and analyzed in 8–15% Tris-Glycine gels and subjected to western blotting with appropriate antibodies: rabbit anti-RBM3 (1/500, 14363-1-AP, Proteintech, Manchester, UK), CIRBP (1/1000, 10209-2-AP, Proteintech), and mouse anti-β-actin (1/5000, SC1615, Santa Cruz, TX, USA). Briefly, proteins were transferred to a nitrocellulose membrane (GE Healthcare) and blocked 1 hr with the appropriate blocking solution. Membranes were incubated overnight at 4°C with the primary antibodies and then with the corresponding donkey anti-mouse and anti-rabbit horseradish peroxidase-conjugated secondary antibodies (Jackson ImmunoResearch, West Grove, PA, USA). All membranes were revealed using ECL chemiluminescence kit according to the manufacturer's instructions (GE Healthcare) and blots were analyzed with ImageJ software (NIH).

## Preparation of acute rat hippocampal slices, electrophysiological recordings, and data analysis

Hippocampal slices (350 µm thick) were prepared from Sprague-Dawley rats (n=3; 3–4 weeks of age; Janvier Laboratories) and cut using a vibratome (Leica VT1200S) in an ice-cold oxygenated, modified ACSF, continuously aerated with 95% $O_2$ and 5% $CO_2$ and containing glucose 11 mM, $NaHCO_3$ 25 mM, NaCl 126 mM, KCl 3.5 mM, $NaH_2PO_4$ 1.2 mM, $MgCl_2$ 1.3 mM, and $CaCl_2$ 2 mM. Slices were then incubated at RT for at least 1 hr in ACSF. Recordings were performed on hippocampal slices using multielectrode arrays (MEA). Slices were continuously perfused with the oxygenated ACSF at the rate of 3 mL/min with a peristaltic pump (MEA chamber volume: ~1 mL). KA (300 nM) and VH-N412 (0.1, 1, and 10 µm) were added to the perfusion solution to assess the effects of VH-N412 compound on KA-induced increase of neuronal firing. Complete solution exchange in the MEA chamber was achieved 20 s after the switch of solutions. The perfusion liquid was continuously preheated at 37°C just before reaching the MEA chamber with a heated-perfusion cannula (PH01, MultiChannel Systems, Reutlingen, Germany). The temperature of the MEA chamber was maintained at 37 ± 0.1°C with a heating element located in the MEA amplifier headstage. The spike numbers per second recorded at each electrode were averaged for 30 s slots and normalized to the mean spikes rate value at t=20–30 min (10 last minutes of KA exposure period). Individual data from independent experiments were then pooled and the mean values of the normalized spike rates (± SEM) were plotted as a function of time (before and after exposure to VH-N412). The control values (KA alone) were averaged from 3 rats, 3 slices, and 18 electrodes. The dose-response curves from the KA+VH-N412-treated slices were averaged from 3 rats, 4 slices, and 25 electrodes.

## Primary cultures of rat hippocampal neurons, transfection, and immunofluorescence

The primary hippocampal cells (mixed culture) were prepared from embryonic day 17–18 (E17 or E18) Wistar rats and cultured in Neurobasal supplemented with 2% B-27, 1% penicillin-streptomycin, and 0.3% glutamine in a humidified atmosphere containing 5% $CO_2$ at 37°C. At 21 DIV neurons displayed mature morphological and physiological features (*Ivanov et al., 2009*). Hippocampal cultures were immunostained for endogenous NTSR1 using goat polyclonal anti-NTSR1 (NTR1, 1/200, R20, Santa Cruz), for a dendritic marker using mouse anti-MAP2 (1/1000, Sigma), and for a spine marker using rabbit anti-drebrin E/A (1/1000, Sigma). In some experiments, neurons were transiently co-transfected with RFP vector (red fluorescent protein) and with a plasmid construct encoding rat NTSR1 using Lipofectamine 2000 reagent according to the manufacturer's protocol (Thermo Fisher Scientific). Cells

were fixed with 4% PFA in PB 0.12 M for 20 min at RT and immunostained for NTSR1 48 hr following transfection. Cell-coated coverslips were rinsed three times with PB 0.12 M and permeabilized in a blocking solution containing 3% BSA and 0.1% Triton X-100 diluted in PB 0.12 M for 30 min at RT. Cells were incubated overnight with primary antibodies diluted in 3% BSA blocking solution inside a humidity chamber at 4°C. Coverslips were washed three times for 5 min in PB 0.12 M, then incubated with corresponding secondary antibodies (Jackson ImmunoReasearch, Cambridgeshire, UK): donkey anti-goat IgG (H+L) highly cross-adsorbed Alexa Fluor A594 (1/800), donkey anti-mouse IgG(H+L) highly cross-adsorbed Alexa Fluor A 647 (1/800), and donkey anti-rabbit IgG (H+L) highly cross-adsorbed Alexa Fluor A 488 (1/800) in PB 0.12 M containing 3% BSA at RT for 2 hr and washed three times for 5 min each with PB 0.12 M. Nuclei were counterstained with 5 µg/mL DAPI at RT for 0.5 hr. After three washes in PB 0.12 M, coverslips were rapidly rinsed three times in distilled water and let to dry before mounting on Superfrost glass slides using Fluoromount-G Mounting medium and stored at –20°C. Labeling specificity was assessed under the same conditions, by incubating some coverslips with non-transfected or transfected cells in a solution omitting the primary antibodies. In all cases, no overlap of antibodies was detected. Image acquisition was performed using Zen software and processed using Adobe Photoshop and ImageJ softwares.

## NMDA intoxication and drug treatment

For the glutamatergic agonist intoxication with NMDA or KA, cells were seeded at the density of 20,000 cells/well in 96-well plates precoated with poly-L-lysine 10 µg/mL and cultured for 21 DIV. On day 21 of culture, the medium was removed, and fresh medium was added without or with test compounds (VH-N412, NTSR antagonists), BDNF (PAN-Biotech, Aidenbach, Germany) or oestradiol (Sigma) 30 min before either NMDA (100 µM, 10 min) or KA (25 µM, 24 hr) intoxication. BDNF (50 ng/mL) and oestradiol (100 nM) were used as positive controls of neuroprotection against NMDA and KA induced injury respectively (*Kajta and Lasoń, 2000*; *Mattson et al., 1995*). Twenty four or 48 hr after NMDA or KA intoxication, medium was removed and cells were washed twice in PB 0.12 M, followed by fixation with 4% PFA (Sigma) for 20 min at RT. Cells were permeabilized and nonspecific sites were blocked with PB 0.12 M containing 0.1% saponin (Sigma) and 1% FCS (Invitrogen) and then incubated with mouse monoclonal anti-MAP-2. This antibody was revealed with Alexa Fluor 488 goat anti-mouse. Cell nuclei were labeled with Hoechst solution (Sigma). Neuronal death was assessed by counting the total number of neuronal cell bodies stained with Hoechst and MAP-2 and measuring the total dendrite length (all neurites stained with MAP-2). For each condition, 10 pictures per well were taken using Developer software and InCell 2000 image system (GE HealthCare, Pittsburgh, PA, USA) with ×20 magnification. All images were taken in the same conditions.

## Statistical analysis

All quantifications were performed blindly. Sample size and statistical power were determined according to *Dell et al., 2002*, and *Festing and Altman, 2002*, using biostaTGV software (http://biostatgv.sentiweb.fr). All experiments were performed at least three times with different series of mice or independent cultures of rat hippocampal neurons. Student's t-test was used to compare two groups. ANOVA followed by Dunnett's or Tukey's post hoc test was used for multiple comparison. Behavioral data were analyzed with MANOVAs with repeated measures followed by Newman-Keuls post hoc comparisons, using the SPSS/PC+ statistics 11.0 software (SPSS Inc, IL, USA) and selected ANOVAs were performed when necessary. All data were expressed as the mean ± SEM. Statistical significance was set at *p<0.05, **p<0.01, and ***p<0.001.

## Acknowledgements

In memoriam: we dedicate this article to the late Professor Jean-Michel Scherrmann, Professor in Pharmacokinetics and Honorary Dean of the Faculty of Pharmacy of Paris.

Financial support was provided by the French National Agency for Research (VECtoBrain ANR-09-BIOT-015-01 to VECT-HORUS, VEC2Brain ANR-13-RPIB-0010-01 to MK and NANOVECTOR ANR-15-CE18-0010-03 to MK), by the CNRS and Aix Marseille Université. We thank the animal facility of the Faculty of Pharmacy of Paris-Cité Université (US25 Inserm, UAR3612 CNRS) for hosting the animals for the in situ brain perfusion experiments.

# Additional information

## Competing interests

Michel Khrestchatisky: Was director of the Institute of Neurophysiopathology, UMR7051 academic neuroscience laboratory supported by the CNRS and Aix-Marseille Université, but also co-founder, shareholder and scientific counsel of the VECT-HORUS biotechnology company. The other authors declare that no competing interests exist.

## Funding

| Funder | Grant reference number | Author |
|---|---|---|
| Agence Nationale de la Recherche | ANR-09-BIOT-015-01 | Jamal Temsamani |
| Agence Nationale de la Recherche | ANR-13-RPIB-0010-01 | Michel Khrestchatisky |
| Agence Nationale de la Recherche | ANR-15-CE18-0010-03 | Michel Khrestchatisky |
| Centre National de la Recherche Scientifique (CNRS) and Aix-Marseille University (AMU) | | Michel Khrestchatisky |

The funders had no role in study design, data collection and interpretation, or the decision to submit the work for publication.

## Author contributions

Lotfi Ferhat, Conceptualization, Data curation, Formal analysis, Supervision, Investigation, Visualization, Methodology, Writing – review and editing; Rabia Soussi, Maxime Masse, Data curation, Formal analysis, Methodology; Grigorios Kyriatzis, Fanny Gassiot, Nicolas Gaudin, Mathieu Laurencin, Anne Bernard, Angélique Bôle, Géraldine Ferracci, Maria Smirnova, Data curation, Formal analysis, Investigation, Methodology; Stéphane Girard, Data curation, Formal analysis, Investigation, Methodology, Writing – original draft; François Roman, Formal analysis, Investigation, Methodology; Vincent Dive, Salvatore Cisternino, Formal analysis, Supervision, Validation, Investigation; Jamal Temsamani, Supervision, Validation, Project administration; Marion David, Supervision, Investigation, Methodology, Writing – original draft; Pascaline Lécorché, Formal analysis, Supervision, Investigation, Methodology, Writing – original draft; Guillaume Jacquot, Formal analysis, Supervision, Validation, Investigation, Writing – original draft; Michel Khrestchatisky, Conceptualization, Resources, Supervision, Funding acquisition, Validation, Methodology, Project administration, Writing – review and editing

## Author ORCIDs

Lotfi Ferhat ⓘ https://orcid.org/0000-0003-2763-9472
Michel Khrestchatisky ⓘ https://orcid.org/0000-0003-4797-8332

## Ethics

All animals were housed 6 per cage in a temperature and humidity-controlled room (22±2°C, 12 hr light-dark cycles), had free access to food and water. Procedures involving animals were carried out according to National and European regulations (EU Directive N°2010/63) and to authorizations delivered to our animal facility (N° C13 055 08) and to the project (N° 00757.02) by the French Ministry of Research and Local Ethics Committee. All efforts were made to minimize animal suffering and reduce the number of animals used.

Reviewer #1 (Public review): https://doi.org/10.7554/eLife.100527.4.sa1
Reviewer #2 (Public review): https://doi.org/10.7554/eLife.100527.4.sa2
Author response https://doi.org/10.7554/eLife.100527.4.sa3

## Additional files

### Supplementary files

Supplementary file 1. Analytical characterization of peptide-neurotensin (NT) conjugates.

Supplementary file 2. Hypothermic response to intravenous (i.v.) administration of peptide-neurotensin (NT) conjugates in mice.

Supplementary file 3. Comparison of mean body temperature changes between SHAM, status epilepticus (SE), SE+VH-N412, SE+neurotensin (NT) (8–13), and SE+diazepam (DZP).

Supplementary file 4. Comparison of mean seizure intensity score between SHAM, status epilepticus (SE), SE+VH-N412, SE+neurotensin (NT) (8–13), and SE+diazepam (DZP).

Supplementary file 5. Mouse and rat TaqMan probes used for qPCR analysis.

MDAR checklist

### Data availability

We provide a complete MDAR checklist and a Key Resources Table. We provide raw data for western blots at https://doi.org/10.5061/dryad.nzs7h451x; raw data for behavior in olfactory tubing maze and open field at https://doi.org/10.5061/dryad.6wwpzgn8k; raw data obtained from multi-electrode electrophysiological recordings of hippocampal slices at https://doi.org/10.5061/dryad.vdncjsz5k; and raw data for hypothermia and seizure intensity scores at https://doi.org/10.5061/dryad.b5mkkwhq1.

The following datasets were generated:

| Author(s) | Year | Dataset title | Dataset URL | Database and Identifier |
|---|---|---|---|---|
| Ferhat L, Soussi R, Masse M, Kyriatzis G, Girard S, Gassiot F, Gaudin N, Laurencin M, Bernard A, Bôle A, Ferracci G, Smirnova M, Roman F, Dive V, Cisternino S, Temsamani J, David M, Lécorché P, Jacquot G, Khrestchatisky M | 2025 | Raw Western blots data to assess cold shock protein expression | https://doi.org/10.5061/dryad.nzs7h451x | Dryad Digital Repository, 10.5061/dryad.nzs7h451x |
| Ferhat L, Soussi R, Masse M, Kyriatzis G, Girard S, Gassiot F, Gaudin N, Laurencin M, Bernard A, Bôle A, Ferracci G, Smirnova M, Roman F, Dive V, Cisternino S, Temsamani J, David M, Lécorché P, Jacquot G, Khrestchatisky M | 2025 | Raw data behavior in olfactory tubing maze and open field | https://doi.org/10.5061/dryad.6wwpzgn8k | Dryad Digital Repository, 10.5061/dryad.6wwpzgn8k |
| Ferhat L, Soussi R, Masse M, Kyriatzis G, Girard S, Gassiot F, Gaudin N, Laurencin M, Bernard A, Bôle A, Ferracci G, Smirnova M, Roman F, Dive V, Cisternino S, Temsamani J, David M, Lécorché P, Jacquot G, Khrestchatisky M | 2025 | Raw data obtained from multi-electrode electrophysiological recordings of hippocampal slices | https://doi.org/10.5061/dryad.vdncjsz5k | Dryad Digital Repository, 10.5061/dryad.vdncjsz5k |

*Continued on next page*

*Continued*

| Author(s) | Year | Dataset title | Dataset URL | Database and Identifier |
|-----------|------|---------------|-------------|--------------------------|
| Ferhat L, Soussi R, Masse M, Kyriatzis G, Girard S, Gassiot F, Gaudin N, Laurencin M, Bernard A, Bôle A, Ferracci G, Smirnova M, Roman F, Dive V, Cisternino S, Temsamani J, David M, Lécorché P, Jacquot G, Khrestchatisky M | 2025 | Raw data of hypothermia and seizure intensity scores | https://doi.org/10.5061/dryad.b5mkkwhq1 | Dryad Digital Repository, 10.5061/dryad.b5mkkwhq1 |

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

# Appendix 1

**Appendix 1—key resources table**

| Reagent type (species) or resource | Designation | Source or reference | Identifiers | Additional information |
|---|---|---|---|---|
| Strain, strain background (*M. musculus*) | Swiss CD-1 mice | Janvier laboratories, France | Cat#: RjOrl:SWISS | Young adult males 5 week old |
| Strain, strain background (*M. musculus*) | FVB/N mice | Janvier laboratories, France | Cat#: FVB/NRj | Adult males 12-week-old |
| Strain, strain background (Rattus) | Sprague Dawley | Janvier laboratories, France | Cat#: RRID:MGI:5651135 | Young adult, 3–4 week old |
| Strain, strain background (Rattus) | Wistar | Janvier laboratories, France | Cat#: RjHan:WI | Young adult, 3–4 week old |
| Cell line (Homo-sapiens) | HEK293 | ECACC, Salisbury, UK | Cat#: RRID:CVCL_0045 | |
| Transfected construct (Homo-sapiens) | Ntsr1 (untagged ORF) - Human neurotensin receptor 1 (Ntsr1) plasmid construct | Origene, Rockville, Maryland, USA | Cat#:SKU SC313374 | |
| Transfected construct (Rattus) | Ntsr1 (untagged ORF) - Rat neurotensin receptor 1 (Ntsr1) plasmid construct | Origene, Rockville, Maryland, USA | Cat#:SKU RN202030 | |
| Biological sample | Human NTSR1 membrane homogenates | This paper | | |
| Biological sample | Rat NTSR1 membrane homogenates | This paper | | |
| Antibody | Neuronal nuclear antigen (NeuN) Clone A60 | Millipore, Darmstadt, Germany | Cat#: RRID:AB_2298772 | Dilution 1:1000 |
| Antibody | Mouse glial fibrillary acidic protein GFAP Clone GA5 | Millipore, Darmstadt, Germany | Cat#: MAB360 | Dilution 1:1000 |
| Antibody | Rabbit anti-ionized calcium binding adaptor molecule 1 Iba1 | Wako Pure Chemical Industries, Osaka, Japan | Cat#: 019–19741 | Dilution 1:1000 |
| Antibody | Rabbit anti-vesicular zinc transporter 3 (ZnT-3) | Synaptic System, Goettingen, Germany | Cat#: RRID:AB_2744656 | Dilution 1:500 |
| Antibody | Alexa 488-conjugated goat anti-rabbit IgG | ThermoFisher Scientific, Waltham, MA, USA | Cat#: RRID:AB_143165 | Dilution 1:200 |
| Antibody | Alexa 488-conjugated goat anti-mouse IgG | ThermoFisher Scientific, Waltham, MA, USA | Cat#: RRID:AB_346865 | Dilution 1:200 |
| Antibody | Rabbit anti-RBM3 | Proteintech, Manchester, UK | Cat: 14363–1-AP | Dilution 1:500 |
| Antibody | Rabbit anti-CIRBP | Proteintech, Manchester, UK | Cat#: 10209–2-AP | Dilution 1:1000 |
| Antibody | Mouse anti-b-actin | Santa-Cruz, TX, USA | Cat#: SC1615 | Dilution 1:5000 |
| Antibody | Horseradish peroxidase-conjugated donkey anti-mouse antibody | Jackson ImmunoResearch, West Grove, PA, USA | Cat#: RRID:AB_2340770 | Dilution 1:10000 |
| Antibody | Horseradish peroxidase-conjugated donkey anti-rabbit antibody | Jackson ImmunoResearch, West Grove, PA, USA | Cat#: RRID:AB_10015282 | Dilution 1:10000 |
| Antibody | Goat polyclonal anti-NTSR1 (NTR1) | Santa-Cruz, TX, USA | Cat#: SC-7598 | Dilution 1:200 |
| Antibody | Mouse anti-MAP2, Clone HM-2 | Sigma-Aldrich (St. Louis, MO, USA) | Cat#: M-4403 | Dilution 1:1000 |
| Antibody | Rabbit anti-Drebrin E/A | Sigma-Aldrich (St. Louis, MO, USA) | Cat#: AB10140 | Dilution 1:1000 |

*Appendix 1 Continued on next page*

*Appendix 1 Continued*

| Reagent type (species) or resource | Designation | Source or reference | Identifiers | Additional information |
|---|---|---|---|---|
| Antibody | Donkey anti-goat IgG highly cross-adsorbed AlexaFluor A594 | Jackson ImmunoResearch, West Grove, PA, USA | Cat#: RRID:AB_2340433 | Dilution 1:800 |
| Sequence-based reagent | TaqMan probe, mouse CIRBP (Cold Inducible RNA Binding Protein) | ThermoFisher Scientific, Waltham, MA, USA | Cat#: Mn00483336 | |
| Sequence-based reagent | TaqMan probe, rat CIRBP (Cold Inducible RNA Binding Protein) | ThermoFisher Scientific, Waltham, MA, USA | Cat#: Rn00579806 | |
| Sequence-based reagent | TaqMan probe, mouse RBM3 (RNA Binding Motif Protein 3) | ThermoFisher Scientific, Waltham, MA, USA | Cat#: Mn00812518 | |
| Sequence-based reagent | TaqMan probe, rat RBM3 (RNA Binding Motif Protein 3) | ThermoFisher Scientific, Waltham, MA, USA | Cat#: Rn01525079 | |
| Sequence-based reagent | TaqMan probe, mouse GAPDH (Glyceraldehyde-3 phosphate dehydrogenase) | ThermoFisher Scientific, Waltham, MA, USA | Cat#: Mn01253033 | |
| Sequence-based reagent | TaqMan probe, rat GAPDH (Glyceraldehyde-3 phosphate dehydrogenase) | ThermoFisher Scientific, Waltham, MA, USA | Cat#: Rn01253033 | |
| Peptide | peptide 22: [cMPRLRGC]$_c$ | This paper | *Malcor et al., 2012* | |
| Peptide | Ac-cMPRLRGC-G-OH | This paper | | |
| Peptide | Ac-[cMPRLRGC]$_c$-G-OH | This paper | | |
| Peptide | pELYENKPRRPYIL-OH | This paper | | |
| Peptide | VH-N21 | This paper | | |
| Peptide | VH04129 [cM"Pip"RLR"Sar"C]$_c$ | This paper | | |
| Peptide | VH-N41 | This paper | | |
| Peptide | VH-N412 | This paper | | |
| Peptide | [$^3$H]-neurotensin | Perkin Elmer | https://resources.perkinelmer.com/lab-solutions/resources/docs/sht_008241d_01.pdf | |
| Recombinant protein | Recombinant human LDLR (His-tagged) | Sino Biological (Beijing, China). | Cat. #: 10,231 H08H | |
| Commercial assay or kit | DC protein assay kit | Bio-Rad, Hercules, CA, USA | Cat. #: 5000111 | |
| Commercial assay or kit | Nucleospin RNA plus kit | Macherey Nagel, Allentown, PA, USA | Cat. #:740984.50 | |
| Commercial assay or kit | ECL chemiluminescence kit | GE Healthcare | Cat. #: 28980926 | |
| Chemical compound | Fmoc-Gly-Wang Resin (100–200 mesh) | Iris Biotech | Cat. #: WAA11313 | |
| Chemical compound | Fmoc-Leu-Wang Resin (100–200 mesh) | Iris Biotech | Cat. #: WAA11316 | |
| Chemical compound | Lipofectamine 2000 | ThermoFisher Scientific, Waltham, MA, USA | Cat. #: 11668027 | |
| Chemical compound | Kainic acid (kainate, KA) | Abcam, Cambridge, UK | Cat. #: ab120100 | |
| Chemical compound | Doliprane (Paracetamol) | Sanofi, Gentilly, France | | Doliprane gel, 2.4% |
| Chemical compound | Diazepam (Valium$^R$) | Roche, Basel, Switzerland | | 5 mg/ml |

*Appendix 1 Continued on next page*

*Appendix 1 Continued*

| Reagent type (species) or resource | Designation | Source or reference | Identifiers | Additional information |
|---|---|---|---|---|
| Chemical compound | Pentobarbital (Nembutal) | Sigma-Aldrich (St. Louis, MO, USA) | Cat. #: 76-74-4 | 700 mg/kg |
| Chemical compound | Fluoro-Jade C (FJC) | Sigma-Aldrich (St. Louis, MO, USA) | Cat#: AG325 | Dilution 1:10000 |
| Chemical compound | BDNF | PAN-Biotech, Aidenbach, Germany | Cat. #:CB-1115001 | |
| Chemical compound | Oestradiol | Sigma-Aldrich (St. Louis, MO, USA) | Cat. #: E9875 | |
| Chemical compound | N-Methyl-D-aspartic acid (NMDA) | Sigma-Aldrich (St. Louis, MO, USA) | Cat. #:6384-92-5 | |
| Software, algorithm | ImageJ softwares | Free access software | https://imagej.net/ij/ | |
| Software, algorithm | LabVIEW software | National Instruments (NI) Austin, TX 78759–3504 USA | https://www.ni.com/fr-fr/shop/product/labview.html?srsltid=AfmBOoo7GoVIL1yty98ffFmJFGqN8m64ST5WMFlMol6lBqQDYAtbLPcs | |
| Software, algorithm | Viewpoint VideoTrack version 3.0 | Lyon, France | https://www.viewpoint.fr/product/rodent/rodents-behavior-monitoring/videotrack | |
| Software, algorithm | 7500 v2.0 software | Applied Biosystems | https://www.thermofisher.com/fr/fr/home/technical-resources/software-downloads/applied-biosystems-7500-real-time-pcr-system.html | |
| Software, algorithm | biostaTGV software | Free access software, | http://biostatgv.sentiweb.fr | |
| Software, algorithm | IBM SPSS Statistics for Windows, Version 20.0. | IBM Corp. Released 2011. Armonk, NY, USA | https://www.ibm.com/support/pages/downloading-ibm-spss-statistics-20 | |
| Software, algorithm | Zen software | Zeiss, Jena Germany | https://www.zeiss.com/microscopy/en/products/software/zeiss-zen.html | |
| Software, algorithm | Developer software used with InCell 2000 image system | GE Healthcare, Pittsburgh, PA, USA | https://download.cytivalifesciences.com/cellanalysis/download_data/incell/incell_docs/IN%20Cell%20Analyzer%202200%20User%20Manual.pdf | |

