## [Editor Report · eLife Assessment]

The authors developed a method to allow a hypothermic agent, neurotensin, to cross the blood-brain barrier so it could potentially protect the brain from seizures and the adverse effects of seizures. The work is **important** because it is known that cooling the brain can protect it but developing a therapeutic approach based on that knowledge has not been done. The paper is well presented and the data are **convincing**.

---

## [Referee Report · Reviewer #1 (Public review)]

In this manuscript, Ferhat and colleagues describe their study aimed at developing a blood brain barrier (BBB) penetrant agent that could induce hypothermia and provide neuroprotection from the sequelae of status epilepticus (SE) in mice. Hypothermia is used clinically in an attempt to reduce neurological sequelae of injury and disease. Hypothermia can be effective, but physical means used to reduce core body temperature is associated with untoward effects. Pharmacological means to induce hypothermia could be as effective with fewer untoward complications. Intracerebroventricularly applied neurotensin can cause hypothermia; however, neurotensin applied peripherally is degraded and does not cross the BBB. Here the authors develop and characterize a neurotensin conjugate that can reach the brain, induce hypothermia, and reduce seizures, cognitive changes, and inflammatory changes associated with status epilepticus.

Strengths:

(1) In general, the study is well reasoned, well designed, and seemingly well executed.

(2) Strong dose-response assessment of multiple neurotensin conjugates in mice.

(3) Solid assessment of binding affinity, in vitro stability ion blood, and brain uptake of the conjugate.

(4) Appropriate inclusion of controls for SE and for drug injections.

(5) Multifaceted assessment of neurodegeneration, inflammation, and mossy fiber sprouting in the different groups.

(6) Inclusion of behavioral assessments.

(7) Evaluate NSTR1 receptor distribution in multiple ways.

(8) Demonstrate that this conjugate can induce hypothermia and have positive effects on the sequelae of SE. Could have great impact on the application of pharmacologically-induced hypothermia as a neuroprotective measure in patients.

Weaknesses:

(1) The data suggest that the neurotensin conjugate causes hypothermia AND has favorable effects on the sequelae of SE. There is a limitation that they do not definitely show that the hypothermia caused by the neurotensin conjugate is necessarily responsible for the effects they see. The authors recognize and discuss this limitation in the manuscript.

---

## [Referee Report · Reviewer #2 (Public review)]

Summary:

The authors generated analogs consisting of modified neurotensin (NT) peptides capable of binding to low density lipoprotein (LDL) and NT receptors. Their lead analog was further evaluated for additional validation as a novel therapeutic. The putative mechanism of action for NT in its antiseizure activity is hypothermia, and as therapeutic hypothermia has been demonstrated in epilepsy, NT analogs may confer antiseizure activity and avoid the negative effects of induced hypothermia.

Strengths:

The authors demonstrate an innovative approach, i.e. using LDLR as a means of transport into the brain, that may extend to other compounds. They systematically validate their approach and its potential through binding, brain penetration, in vivo antiseizure efficacy, and neuroprotection studies.

---

## [Author Response]

The following is the authors’ response to the previous reviews.

We addressed the issue of “tolerability” in our answers to Reviewer 2 and in the revised manuscript where we had added data concerning tolerability, see the paragraph in the Results Section, page 11:

"Finally, tolerability studies were performed with the administration of up to 20 and 40 mg/kg eq. NT (i.e. 25.8 and 51.6 mg/kg of VH-N412) with n=3 for these doses. The rectal temperature of the animals did not fall below 32.5 to 33.2°C, similar to the temperature induced with the 4 mg/kg eq. NT dose. We observed no mortality or notable clinical signs other than those associated with the rapid HT effect such as a decrease in locomotor activity. We thus report a very interesting therapeutic index since the maximal tolerated dose (MTD) was > 40 mg/kg eq. NT, while the maximum effect is observed at a 10x lower dose of 4 mg/kg eq. NT and an ED50 established at 0.69 mg/kg as shown in Figure 1G.”

We have slightly modified the paragraph above to emphasize that the tolerability studies were performed in “naïve mice”.

"Finally, tolerability studies were performed in naïve mice with the administration of up to 20 and 40 mg/kg eq. NT (i.e. 25.8 and 51.6 mg/kg of VH-N412) with n=3 for these doses. The rectal temperature of the animals did not fall below 32.5 to 33.2°C, similar to the temperature induced with the 4 mg/kg eq. NT dose. We observed no mortality or notable clinical signs other than those associated with the rapid HT effect such as a decrease in locomotor activity. We thus report a very interesting therapeutic index since the maximal tolerated dose (MTD) was > 40 mg/kg eq. NT, while the maximum effect is observed at a 10x lower dose of 4 mg/kg eq. NT and an ED50 established at 0.69 mg/kg as shown in Figure 1G.”

We propose to add a sentence in the Results section, page 11, relative to the fact that we can also induce severe hypothermia in rats using conjugates similar to VH-N412.

We also added in the Discussion section (page 38) that we could induce hypothermia with different conjugates in mice, rats and pigs.